# Bayesian Optimization over Discrete and Mixed Spaces via Probabilistic Reparameterization

**Samuel Daulton**
University of Oxford, Meta
sdaulton@meta.com

**Xingchen Wan**
University of Oxford
xwan@robots.ox.ac.uk

**David Eriksson**
Meta
deriksson@meta.com

**Maximilian Balandat**
Meta
balandat@meta.com

**Michael A. Osborne**
University of Oxford
mosb@robots.ox.ac.uk

**Eytan Bakshy**
Meta
ebakshy@meta.com

## Abstract

Optimizing expensive-to-evaluate black-box functions of discrete (and potentially continuous) design parameters is a ubiquitous problem in scientific and engineering applications. Bayesian optimization (BO) is a popular, sample-efficient method that leverages a probabilistic surrogate model and an acquisition function (AF) to select promising designs to evaluate. However, maximizing the AF over mixed or high-cardinality discrete search spaces is challenging standard gradient-based methods cannot be used directly or evaluating the AF at every point in the search space would be computationally prohibitive. To address this issue, we propose using probabilistic reparameterization (PR). Instead of directly optimizing the AF over the search space containing discrete parameters, we instead maximize the expectation of the AF over a probability distribution defined by continuous parameters. We prove that under suitable reparameterizations, the BO policy that maximizes the probabilistic objective is the same as that which maximizes the AF, and therefore, PR enjoys the same regret bounds as the original BO policy using the underlying AF. Moreover, our approach provably converges to a stationary point of the probabilistic objective under gradient ascent using scalable, unbiased estimators of both the probabilistic objective and its gradient. Therefore, as the number of starting points and gradient steps increase, our approach will recover of a maximizer of the AF (an often-neglected requisite for commonly used BO regret bounds). We validate our approach empirically and demonstrate state-of-the-art optimization performance on a wide range of real-world applications. PR is complementary to (and benefits) recent work and naturally generalizes to settings with multiple objectives and black-box constraints.

## 1 Introduction

Many scientific and engineering problems involve tuning discrete and/or continuous parameters to optimize an objective function. Often, the objective function is "black-box", meaning it has no known closed-form expression. For example, optimizing the design of an electrospun oil sorbent—a material that can be used to absorb oil in the case of a marine oil spill to mitigate ecological harm—to maximize properties such as the oil absorption capacity and mechanical strength [59] can involve tuning both discrete ordinal experimental conditions and continuous parameters controlling the composition of the material. For another example, optimizing the structural design of a welded beam can involve tuning the type of metal (categorical), the welding type (binary), and the dimensions of the different components of the beam (discrete ordinals)–resulting in a search space with over 370 million possible designs [54]. We consider the scenario where querying the objective function is

expensive and sample-efficiency is crucial. In the case of designing the oil sorbent, evaluating the objective function requires manufacturing the material and measuring its properties in a laboratory, requiring significant time and resources.

Bayesian optimization (BO) is a popular technique for sample-efficient black-box optimization, due to its proven performance guarantees in many settings [5, 52] and its strong empirical performance [23, 55]. BO leverages a probabilistic surrogate model of the unknown objective(s) and an acquisition function (AF) that provides utility values for evaluating a new design to balance exploration and exploitation. Typically, the maximizer of the AF is selected as the next design to evaluate. However, maximizing the AF over mixed search spaces (i.e., those consisting of discrete and continuous parameters) or large discrete search spaces is challenging[1] and continuous (or gradient-based) optimization routines cannot be directly applied. Theoretical performance guarantees of BO policies require that the maximizer of the AF is found and selected as the next design to evaluate on the black-box objective function [52]. When the maximizer is not found, regret properties are not guaranteed, and the performance of the BO policy may degrade.

To tackle these challenges, we propose a technique for improving AF optimization using a probabilistic reparameterization (PR) of the discrete parameters. Our main contributions are:

1. We propose a technique, probabilistic reparameterization (PR), for maximizing AFs over discrete and mixed spaces by instead optimizing a probabilistic objective (PO): the expectation of the AF over a probability distribution of discrete random variables corresponding to the discrete parameters.

2. We prove that there is an equivalence between the maximizers of the acquisition function and the the maximizers of the PO and hence, the policy that chooses designs that are best with respect to the PO enjoys the same performance guarantees as the standard BO policy.

3. We derive scalable, unbiased Monte Carlo estimators of the PO and its gradient with respect to the parameters of the introduced probability distribution. We show that stochastic gradient ascent using our gradient estimator is guaranteed to converge to a stationary point on the PO surface and will recover a global maximum of the underlying AF as the number of starting points and gradient steps increase. This is important because many BO regret bounds require maximizing the AF [52]. Although the AF is often non-convex and maximization is hard, empirically, with a modest number of starting points, PR leads to better AF optimization than alternative methods.

4. We show that PR yields state-of-the-art optimization performance on a wide variety of real-world design problems with discrete and mixed search spaces. Importantly, PR is *complementary* to many existing approaches such as popular multi-objective, constrained, and trust region-based approaches; in particular, PR is agnostic to the underlying probabilistic model over discrete parameters—which is not the case for many alternative methods.

## 2   Preliminaries

**Bayesian Optimization**   We consider the problem of optimizing a black-box function $f : \mathcal{X} \times \mathcal{Z} \to \mathbb{R}$ over a compact search space $\mathcal{X} \times \mathcal{Z}$, where $\mathcal{X} = \mathcal{X}^{(1)} \times \cdots \times \mathcal{X}^{(d)}$ is the domain of the $d \geq 0$ continuous parameters ($x^{(i)} \in \mathcal{X}^{(i)}$ for $i = 1, ..., d$) and $\mathcal{Z} = \mathcal{Z}^{(1)} \times \cdots \times \mathcal{Z}^{(d_z)}$ is the domain of the $d_z \geq 1$ discrete parameters ($z^{(i)} \in \mathcal{Z}^{(i)}$ for $i = 1, ..., d_z$).[2]

BO leverages (i) a probabilistic surrogate model—typically a Gaussian process (GP) [46]—fit to a data set $\mathcal{D}_n = \{\boldsymbol{x}_i, \boldsymbol{z}_i, y_i\}_{i=1}^n$ of designs and corresponding (potentially noisy) observations $y_i = f(\boldsymbol{x}_i, \boldsymbol{z}_i) + \epsilon_i, \epsilon_i \sim \mathcal{N}(0, \sigma^2)$, and (ii) an acquisition function $\alpha(\boldsymbol{x}, \boldsymbol{z})$ that uses the surrogate model's posterior distribution to quantify the value of evaluating a new design. Common AFs include expected improvement (EI) [32] and upper confidence bound (UCB) [52]—the latter of which enjoys no-regret guarantees in certain settings [52]. The next design to evaluate is chosen by maximizing

---

[1]If the discrete search space has low enough cardinality that the AF can be evaluated at every discrete element, then acquisition optimization can be solved trivially.

[2]Throughout this paper, we use a mixed search space $\mathcal{X} \times \mathcal{Z}$ in our derivations, theorems, and proofs, without loss of generality with respect to the case of a purely discrete search space. If $d = 0$, then the objective function $f : \mathcal{Z} \to \mathbb{R}$ is defined over the discrete space $\mathcal{Z}$ and the continuous parameters in this exposition can simply be ignored.

the AF $\alpha(\boldsymbol{x}, \boldsymbol{z})$ over $\mathcal{X} \times \mathcal{Z}$. Although the black-box objective $f$ is expensive-to-evaluate, the AF is relatively cheap-to-query, and therefore, it can be optimized numerically. Gradient-based optimization routines are often used to maximize the AF over continuous domains [25].

**Discrete Parameters** In its basic form, BO assumes that the inputs are continuous. However, discrete parameters such as binary, discrete ordinal, and non-ordered categorical parameters are ubiquitous in many applications. In the presence of such parameters, optimizing the AF is more difficult, as standard gradient-based approaches cannot be directly applied. Recent works have proposed various approaches including multi-armed bandits [40, 48] and local search [41] for discrete domains and interleaved discrete/continuous optimization procedures for mixed domains [15, 57]. A simple and widely-used approach across many popular BO packages [1, 53] is to one-hot encode the categorical parameters, apply a continuous relaxation when solving the optimization, and discretize (round) the resulting continuous candidates. Examples of continuous relaxations and discretization functions are listed in Table 1.

Table 1: Different parameter types, their continuous relaxations, and discretization functions.

| TYPE | DOMAIN | CONT. RELAXATION | discretize$(\cdot)$ FUNCTION |
|------|--------|------------------|------------------------------|
| BINARY | $z \in \{0, 1\}$ | $z' \in [0, 1]$ | round$(z')$ |
| ORDINAL | $z \in \{0, ..., C - 1\}$ | $z' \in [-0.5, C - 0.5)$ | round$(z')$ |
| CATEGORICAL | $z \in \{0, ..., C - 1\}$ | $z' \in [0, 1]^C$ | $\arg\max_c z'^{(c)}$ |

Although using a continuous relaxation allows for efficient optimization using standard optimization routines in an alternate continuous domain $\mathcal{Z}' \subset \mathbb{R}^m$, the AF value for an infeasible continuous value (i.e., $z' \notin \mathcal{Z}$) does not account for the discretization that must occur before the black-box function is evaluated. Moreover, the acquisition value for an infeasible continuous value can be larger than the AF value after discretization. For an illustration of this, see Fig. 1 (middle/right). In the worst case, BO will repeatedly select the same infeasible continuous design due to its high AF value, but discretization will result in a design that has already been evaluated and has zero AF value. To mitigate this degenerate behavior and avoid the over-estimation issue, Garrido-Merchán and Hernández-Lobato [26] propose discretizing $z'$ before evaluating the AF, but the AF is then non-differentiable with respect to the $z'$. While this improves performance on small search spaces, the response surface has large flat regions after discretizing $z'$, which makes it difficult to optimize the AF. The authors of [26] propose to approximate the gradients using finite differences, but, empirically, we find that this approach to be leads to sub-optimal AF optimization relative to PR.

## 3 Probabilistic Reparameterization

We propose an alternative approach based on probabilistic reparameterization, a relaxation of the original optimization problem involving discrete parameters. Rather than directly optimizing the AF via a continuous relaxation $\boldsymbol{z}'$ of the design $\boldsymbol{z}$, we instead reparameterize the optimization problem by introducing a discrete probability distribution $p(\boldsymbol{Z}|\boldsymbol{\theta})$ over a random variable $\boldsymbol{Z}$ with support exclusively over $\mathcal{Z}$. This distribution is parameterized

---

**Algorithm 1** BO with PR

1: Input: black-box objective $f : \mathcal{X} \times \boldsymbol{Z} \to \mathbb{R}$
2: Initialize $\mathcal{D}_0 \leftarrow \emptyset$, $\text{GP}_0 \leftarrow \text{GP}(\mathbf{0}, k)$
3: **for** $n = 1$ **to** $N_{\text{iterations}}$ **do**
4: $\quad (\boldsymbol{x}_n, \boldsymbol{\theta}_n) \leftarrow \arg\max_{(\boldsymbol{x}, \boldsymbol{\theta}) \in \mathcal{X} \times \Theta} \mathbb{E}_{\boldsymbol{Z} \sim p(\boldsymbol{Z}|\boldsymbol{\theta})}[\alpha(\boldsymbol{x}, \boldsymbol{Z})]$
5: $\quad$ Sample $\boldsymbol{z}_n \sim p(\boldsymbol{Z}|\boldsymbol{\theta}_n)$
6: $\quad$ Evaluate $f(\boldsymbol{x}_n, \boldsymbol{z}_n)$
7: $\quad \mathcal{D}_n \leftarrow \mathcal{D}_{n-1} \cup \{(\boldsymbol{x}_n, \boldsymbol{z}_n, \boldsymbol{f}(\boldsymbol{x}_n, \boldsymbol{z}_n))\}$
8: $\quad$ Update posterior $\text{GP}_n$ given $\mathcal{D}_n$
9: **end for**

---

by a vector of continuous parameters $\boldsymbol{\theta}$. We use $\boldsymbol{z}$ to denote the vector $(z^{(1)}, ..., z^{(d_z)})$, where each element is a different (possibly vector-valued) discrete parameter. Given this reparameterization, we define the probabilistic objective (PO):

$$\mathbb{E}_{\boldsymbol{Z} \sim p(\boldsymbol{Z}|\boldsymbol{\theta})}[\alpha(\boldsymbol{x}, \boldsymbol{Z})]. \tag{1}$$

Algorithm 1 outlines BO with probabilistic reparameterization.

PR allows us to optimize $\boldsymbol{\theta}$ and $\boldsymbol{x}$ over a continuous space to maximize the PO instead of optimizing $\boldsymbol{x}$ and $\boldsymbol{z}$ to maximize $\alpha$ directly over the mixed search space $\mathcal{X} \times \mathcal{Z}$. As we will show later, maximizing

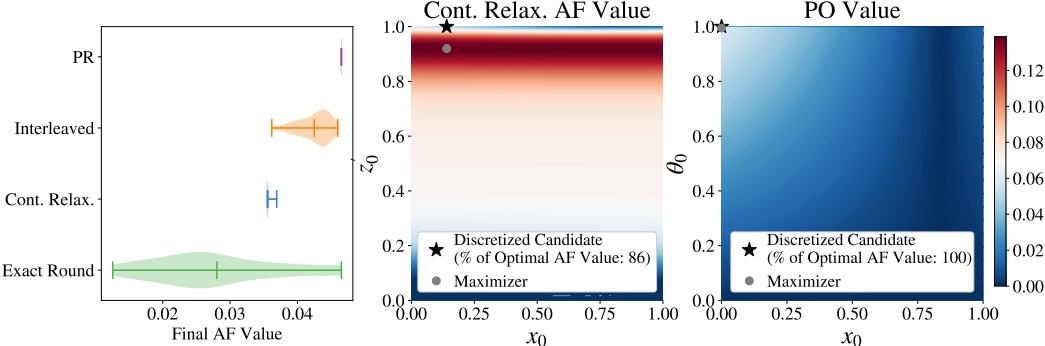

Figure 1: (**Left**) A comparison of AF optimization using different methods over a mixed search space shows that PR *outperforms alternative methods for AF optimization and has much lower variance across replications*. The violin plots show the distribution of final AF values and the mean. "Cont. Relax." denotes optimizing a continuous relaxation of the categoricals with exact gradients. "Exact Round" refers to optimizing a continuous relaxation with approximate gradients (via finite difference), but discretizes the relaxation before evaluating the surrogate [26]. "Interleaved" alternates between one step of local search on the discrete parameters and one step of gradient ascent on the continuous parameters (used in CASMOPOLITAN [57]). For each method, the best candidate across 20 restarts is selected (after discretization) and the acquisition value of the resulting feasible candidate is recorded. The AF is expected improvement [32]. (**Middle/Right**) AF values with a continuous relaxation (middle) and the PO (right) for the Branin function over a mixed domain with one continuous parameter ($x_0$) and one binary parameter ($z_0$) (see Appendix C for details on Branin). (**Middle**) Under a continuous relaxation, the maximizer of the AF is an infeasible point in the domain (grey circle), which results in a suboptimal AF value when rounded (black star); the resulting candidate only has 86% of the AF value of the true maximizer. The maximum AF value across the feasible search space is shown in white and the red regions indicate that the continuous relaxation overestimates the AF value since it is greater than the maximum AF value of any feasible design. (**Right**) The PO is maximized at the AF unique maximizer within the valid search domain. These contours show that PR avoids the overestimation issue that the naive continuous relaxation suffers from.

the PO allows us to recover a maximizer of $\alpha$ over the space $\mathcal{X} \times \mathcal{Z}$. Choosing $p(\boldsymbol{Z}|\theta)$ to be a discrete distribution over $\mathcal{Z}$ means the realizations of $\boldsymbol{Z}$ are feasible values in $\mathcal{Z}$. Hence, the AF is only evaluated for feasible discrete designs. Since $p(\boldsymbol{Z}|\boldsymbol{\theta})$ is a discrete probability distribution, we can express $\mathbb{E}_{\boldsymbol{Z} \sim p(\boldsymbol{Z}|\boldsymbol{\theta})}[\alpha(\boldsymbol{x}, \boldsymbol{Z})]$ as a linear combination where each discrete design is weighted by its probability mass:

$$\mathbb{E}_{\boldsymbol{Z} \sim p(\boldsymbol{Z}|\boldsymbol{\theta})}[\alpha(\boldsymbol{x}, \boldsymbol{Z})] = \sum_{\boldsymbol{z} \in \mathcal{Z}} p(\boldsymbol{z}|\boldsymbol{\theta})\alpha(\boldsymbol{x}, \boldsymbol{z}). \quad (2)$$

Example distributions for binary, ordinal, and categorical parameters are provided in Table 2.

Table 2: Examples of probabilistic reparameterizations for different parameter types. We denote the $(C-1)$-simplex as $\Delta^{C-1}$.

| PARAMETER TYPE | RANDOM VARIABLE | CONTINUOUS PARAMETER |
|---|---|---|
| BINARY | $Z \sim \text{BERNOULLI}(\theta)$ | $\theta \in [0, 1]$ |
| ORDINAL | $Z = \lfloor \theta \rfloor + B, B \sim \text{BERNOULLI}(\theta - \lfloor \theta \rfloor)$ | $\theta \in [0, C-1]$ |
| CATEGORICAL | $Z \sim \text{CATEGORICAL}(\theta), \theta = (\theta^{(1)}, ..., \theta^{(C)})$ | $\theta \in \Delta^{C-1}$ |

Although ordinal parameters could use the same categorical distributions as the non-ordered categorical parameters, we opt for the provided proposal distribution since it uses a scalar $\theta$ (rather than a $C$-element vector) and it naturally encodes the ordering of the values. Using an independent random variable $Z^{(i)} \sim p(Z^{(i)}|\theta^{(i)})$ for each parameter $z^{(i)}$ for $i = 1, ..., d_z$ means that the probabilistic

objective can be expressed as

$$\mathbb{E}_{\boldsymbol{Z} \sim p(\boldsymbol{Z}|\boldsymbol{\theta})}[\alpha(\boldsymbol{x}, \boldsymbol{Z})] = \sum_{z^{(1)} \in \mathcal{Z}^{(1)}} \cdots \sum_{z^{(d_z)} \in \mathcal{Z}^{(d_z)}} \alpha\big(\boldsymbol{x}, z^{(1)}, ..., z^{(d_z)}\big) \prod_{i=1}^{d_z} p(z^{(i)}|\theta^{(i)}). \qquad (3)$$

### 3.1 Analytic Gradients

One important benefit of PR is that the PO in (1) is differentiable with respect to $\boldsymbol{\theta}$ (and $\boldsymbol{x}$, if the gradient of $\alpha$ with respect to $\boldsymbol{x}$ exists), whereas $\alpha(\boldsymbol{x}, \boldsymbol{z})$ is not differentiable with respect to $\boldsymbol{z}$. The gradients of the PO with respect to $\boldsymbol{\theta}$ and $\boldsymbol{x}$ can be obtained by differentiating Equation 2:

$$\nabla_{\boldsymbol{\theta}} \mathbb{E}_{\boldsymbol{Z} \sim p(\boldsymbol{Z}|\boldsymbol{\theta})}[\alpha(\boldsymbol{x}, \boldsymbol{Z})] = \sum_{\boldsymbol{z} \in \mathcal{Z}} \alpha(\boldsymbol{x}, \boldsymbol{z}) \nabla_{\boldsymbol{\theta}} p(\boldsymbol{z}|\boldsymbol{\theta}) \qquad (4)$$

$$\nabla_{\boldsymbol{x}} \mathbb{E}_{\boldsymbol{Z} \sim p(\boldsymbol{Z}|\boldsymbol{\theta})}[\alpha(\boldsymbol{x}, \boldsymbol{Z})] = \sum_{\boldsymbol{z} \in \mathcal{Z}} p(\boldsymbol{z}|\boldsymbol{\theta}) \nabla_{\boldsymbol{x}} \alpha(\boldsymbol{x}, \boldsymbol{z}) \qquad (5)$$

This enables optimizing the PO (line 4 of Algorithm 1) efficiently and effectively using gradient-based methods.

### 3.2 Theoretical Properties

In this section, we derive theoretical properties of PR. Proofs are provided in Appendix B. Our first result is that there is an equivalence between the maximizers of the PO and the maximizers of the AF over $\mathcal{X} \times \mathcal{Z}$.

**Theorem 1** (Consistent Maximizers). *Suppose that $\alpha$ is continuous in $\boldsymbol{x}$ for every $\boldsymbol{z} \in \mathcal{Z}$. Let $\mathcal{H}^*$ be the maximizers of $\alpha(\boldsymbol{x}, \boldsymbol{z})$: $\mathcal{H}^* = \{(\boldsymbol{x}, \boldsymbol{z}) \in \arg \max_{(\boldsymbol{x}, \boldsymbol{z}) \in \mathcal{X} \times \mathcal{Z}} \alpha(\boldsymbol{x}, \boldsymbol{z})\}$. Let $\mathcal{J}^* \subseteq \mathcal{X} \times \Theta$ be the maximizers of $\mathbb{E}_{\boldsymbol{Z} \sim p(\boldsymbol{Z}|\boldsymbol{\theta})}[\alpha(\boldsymbol{x}, \boldsymbol{Z})]$: $\mathcal{J}^* = \{(\boldsymbol{x}, \boldsymbol{\theta}) \in \arg \max_{(\boldsymbol{x}, \boldsymbol{\theta}) \in \mathcal{X} \times \Theta} \mathbb{E}_{\boldsymbol{Z} \sim p(\boldsymbol{Z}|\boldsymbol{\theta})}[\alpha(\boldsymbol{x}, \boldsymbol{Z})]\}$, where $\Theta$ is the domain of $\boldsymbol{\theta}$. Let $\hat{\mathcal{H}}^* \subseteq \mathcal{X} \times \mathcal{Z}$ be defined as: $\hat{\mathcal{H}}^* = \{(\boldsymbol{x}, \tilde{\boldsymbol{z}}) : (\boldsymbol{x}, \boldsymbol{\theta}) \in \mathcal{J}^*, \tilde{\boldsymbol{z}} \sim p(\boldsymbol{Z}|\boldsymbol{\theta})\}$. Then, $\hat{\mathcal{H}}^* = \mathcal{H}^*$.*

Algorithm 1 outlines BO with probabilistic reparameterization. Importantly, Theorem 1 states that sampling from the distribution parameterized by a maximizer of the PO yields a maximizer of $\alpha$, and therefore, Algorithm 1 enjoys the performance guarantees of $\alpha(\cdot)$.

**Corollary 1** (Regret Bounds). *Let $\alpha(\boldsymbol{x}, \boldsymbol{z})$ be an acquisition function over a search space $\mathcal{X} \times \mathcal{Z}$ such that when $\alpha$ is applied as part of a BO strategy that strategy has bounded regret . If the conditions for the regret bounds of that BO strategy using $\alpha$ are satisfied, then Algorithm 1 using $\alpha$ enjoys the same regret bound.*

Examples of BO policies with bounded regret include those based on AFs such as upper confidence bound (UCB) [52] or Thompson sampling (TS) [49] for single objective optimization, and UCB or TS with Chebyshev [43] or hypervolume [66] scalarizations in the multi-objective setting.

Although the BO policy selects a maximizer of $\alpha$ is equivalent to the BO policy in Algorithm 1, maximizing the AF over mixed or high-dimensional discrete search spaces is challenging because commonly used gradient-based methods cannot directly be applied. The key advantage of our approach is that maximizers of the AF can be identified efficiently and effectively by optimizing the PO using gradient information instead of directly optimizing the AF. We find that optimizing PR yields better results than directly optimizing $\alpha$ or other common relaxations as shown in Figure 1 (Left), where we compare AF optimization methods on the mixed Rosenbrock test problem (see Appendix C for details).

## 4 Practical Monte Carlo Estimators

### 4.1 Unbiased estimators of the Probabilistic Reparameterization and its Gradient

As the number of discrete configurations ($|\mathcal{Z}|$) increases, the PO and its gradient may become computationally expensive to evaluate analytically because both require a summation of $|\mathcal{Z}|$ terms. Therefore, we propose to estimate the PO and its gradient using Monte Carlo (MC) sampling. The MC estimator of the PO is given by

$$\mathbb{E}_{\boldsymbol{Z}\sim p(\boldsymbol{Z}|\boldsymbol{\theta})}[\alpha(\boldsymbol{x},\boldsymbol{Z})] \approx \frac{1}{N}\sum_{i=1}^{N}\alpha(\boldsymbol{x},\tilde{\boldsymbol{z}}_i), \tag{6}$$

where $\tilde{\boldsymbol{z}}_1,...,\tilde{\boldsymbol{z}}_N$ are samples from $p(\boldsymbol{Z}|\boldsymbol{\theta})$. This estimator is unbiased and can be computed for a large number of samples by evaluating the AF independently (or in chunks) for each input $(\boldsymbol{x},\tilde{\boldsymbol{z}}_n)$.

MC can also be used to estimate the gradient of the PO with respect to $\boldsymbol{\theta}$. We opt for using a score function gradient estimator [35] (also known as REINFORCE [62] and the likelihood ratio estimator [27]) because it is simple, scalable, and can be computed using the acquisition values $\{\alpha(\boldsymbol{x},\tilde{\boldsymbol{z}}_i)\}_{i=1}^{N}$ that are used in the MC estimator of the PO. Many alternative lower variance estimators (e.g. Yin et al. [64], Yin et al. [65]) would require many additional AF evaluations (see Mohamed et al. [39] for a review of MC gradient estimation). The score function is the gradient of the log probability with respect to the parameters of the distribution: $\nabla_{\boldsymbol{\theta}}\log p(\boldsymbol{Z}|\boldsymbol{\theta}) = \frac{\nabla_{\boldsymbol{\theta}}p(\boldsymbol{Z}|\boldsymbol{\theta})}{p(\boldsymbol{Z}|\boldsymbol{\theta})}$. Using this score function, we can express the analytic gradient as

$$\nabla_{\boldsymbol{\theta}}\mathbb{E}_{\boldsymbol{Z}\sim p(\boldsymbol{Z}|\boldsymbol{\theta})}[\alpha(\boldsymbol{x},\boldsymbol{Z})] = \sum_{\boldsymbol{z}\in\mathcal{Z}}\alpha(\boldsymbol{x},\boldsymbol{z})p(\boldsymbol{z}|\boldsymbol{\theta})\nabla_{\boldsymbol{\theta}}\log p(\boldsymbol{z}|\boldsymbol{\theta}) = \mathbb{E}_{\boldsymbol{Z}\sim p(\boldsymbol{Z}|\boldsymbol{\theta})}[\alpha(\boldsymbol{x},\boldsymbol{Z})\nabla_{\boldsymbol{\theta}}\log p(\boldsymbol{Z}|\boldsymbol{\theta})].$$

The unbiased MC estimator of the gradient of the PO with respect to $\boldsymbol{\theta}$ is given by

$$\nabla_{\boldsymbol{\theta}}\mathbb{E}_{\boldsymbol{Z}\sim p(\boldsymbol{Z}|\boldsymbol{\theta})}[\alpha(\boldsymbol{x},\boldsymbol{Z})] \approx \frac{1}{N}\sum_{i=1}^{N}\alpha(\boldsymbol{x},\tilde{\boldsymbol{z}}_i)\nabla_{\boldsymbol{\theta}}\log p(\tilde{\boldsymbol{z}}_i|\boldsymbol{\theta}). \tag{7}$$

Since the score function gradient is only defined when $p(\boldsymbol{z}|\boldsymbol{\theta}) > 0$, we reparameterize $\boldsymbol{\theta}$ to ensure $p(\boldsymbol{z}|\boldsymbol{\theta}) > 0$ for all $\boldsymbol{z}$ and $\boldsymbol{\theta}$ by using the softmax transformations provided in Table 3, which are commonly used for computational convenience and stability in probablistic reparameterization [64, 65], and the solution converges as $\tau \to 0$.

Moreover, even though $p(\boldsymbol{z}|\boldsymbol{\theta}) > 0$, when $p(\boldsymbol{z}|\boldsymbol{\theta})$ is small, a small number $N$ of MC samples are unlikely to produce any samples where $\tilde{\boldsymbol{z}} = \boldsymbol{z}$. Instead of optimizing $\boldsymbol{\theta}$ directly, we instead optimize $\boldsymbol{\phi}$. Since the transformations $g(\cdot)$ are differentiable with respect to $\boldsymbol{\phi}$, the gradient (and MC gradient estimator) of the PO with respect to $\boldsymbol{\phi}$ are easily obtained using the gradient of the PO with respect to $\boldsymbol{\theta}$ and a simple application the chain rule (multiplying by $\nabla_{\boldsymbol{\phi}}\boldsymbol{\theta}$).

| PARAMETER TYPE | TRANSFORMATION ($\theta = g(\phi)$) |
|---|---|
| BINARY | $\theta = \sigma((\phi - \frac{1}{2})/\tau)$ |
| ORDINAL | $\theta = \lfloor\phi\rfloor + \sigma((\phi - \lfloor\phi\rfloor - \frac{1}{2})/\tau)$ |
| CATEGORICAL | $\theta^{(c)} = \text{SOFTMAX}((\boldsymbol{\phi} - 0.5)/\tau)^{(c)}$ |

Table 3: Transformations where $\tau \in \mathbb{R}_+$ and $\phi, \theta \in \Theta$.

## 4.2 Variance Reduction in Monte Carlo Gradient Estimation

Although the MC gradient estimator in (7) is unbiased, score function gradient estimators can suffer from high variance [39]. Therefore, we adopt a popular technique for variance reduction where the score function itself is used as a control variate, since its expectation is zero under $p(\boldsymbol{Z}|\boldsymbol{\theta})$ [39]. Score function estimators with this control variate have been shown to be among the best performing gradient estimators [39]. Moreover, this technique is simple and merely amounts to subtracting a value $\beta$ from the acquisition value in the score function estimator in Equation (7):

$$\nabla_{\boldsymbol{\theta}}\mathbb{E}_{\boldsymbol{Z}\sim p(\boldsymbol{Z}|\boldsymbol{\theta})}[\alpha(\boldsymbol{x},\boldsymbol{Z})] \approx \frac{1}{N}\sum_{i=1}^{N}[\alpha(\boldsymbol{x},\tilde{\boldsymbol{z}}_i) - \beta]\nabla_{\boldsymbol{\theta}}\log p(\tilde{\boldsymbol{z}}_i|\boldsymbol{\theta}). \tag{8}$$

The $\beta$ is commonly known as a baseline and is often taken to be a moving average of the (acquisition) values [39]. See Appendix C for details on $\beta$.

## 4.3 Convergence Guarantee using Stochastic Gradient Ascent

Since the score function gradient estimator is unbiased, we can leverage previous work on convergence in probability under stochastic gradient ascent [47] to arrive at our main convergence result for acquisition optimization.

**Theorem 2** (Convergence Guarantee). *Let $\alpha : \mathcal{X} \times \mathcal{Z} \to \mathbb{R}$ be differentiable in $\boldsymbol{x}$ for every $\boldsymbol{z} \in \mathcal{Z}$. Let $(\hat{\boldsymbol{x}}_{t,m}, \hat{\boldsymbol{\theta}}_{t,m})$ be the best solution after running stochastic gradient ascent for $t$ time steps on the*

*probabilistic objective $\mathbb{E}_{\boldsymbol{Z}\sim p(\boldsymbol{Z}|\boldsymbol{\theta})}[\alpha(\boldsymbol{x},\boldsymbol{Z})]$ from $m$ starting points with its unbiased MC estimators proposed above. Let $\{a_t\}_{t=1}^{\infty}$ be a sequence of positive step sizes such that $0 < \sum_{t=1}^{\infty} a_t^2 = A < \infty$ and $\sum_{t=1}^{\infty} a_t = \infty$, where $a_t$ is the step size used in stochastic gradient ascent at time step $t$. Let $\hat{\boldsymbol{z}}_{t,m} \sim p(\boldsymbol{Z}|\hat{\boldsymbol{\theta}}_{t,m})$. Then as $t \to \infty$, $m \to \infty$, and $\tau \to 0$, $(\hat{\boldsymbol{x}}_{t,m}, \hat{\boldsymbol{z}}_{t,m}) \to (\boldsymbol{x}^*, \boldsymbol{z}^*) \in \arg\max_{(\boldsymbol{x},\boldsymbol{z})\in\mathcal{X}\times\mathcal{Z}} \alpha(\boldsymbol{x},\boldsymbol{z})$ in probability.*

The significance of Theorem 2 is that optimizing the PO is guaranteed to converge in probability to a global maximizer of the AF, meaning that optimizing the PO guarantees that resulting candidate design has maximal AF value. The implication is that the intended BO policy is followed and the underlying regret bounds of the AF are recovered (provided that the other conditions of the regret bound are met). Although global convergence is only guaranteed as $m \to \infty$, we observe in Figure 1(left) that PR yields strong, stable acquisition optimization with only $m = 20$ starting points, 200 steps, and $\tau = \frac{1}{10}$ (see Appendix G for further discussion) and outperforms alternative optimization approaches.

## 5 Related Work

Many methods for BO over discrete and mixed search spaces have been proposed. Previous work has largely focused on (i) improving the surrogate models or (ii) improving AF optimization.

**Improving models**: Historically, methods leveraging tree-based surrogate models, e.g., SMAC [30] and TPE [4], have been popular for optimizing discrete or mixed search spaces. Many recent works have considered alternative surrogate models. BOCS encodes categorical parameters as binary variables and uses Bayesian linear regression with pairwise interactions [2]. COMBO uses a diffusion kernel on the graph defined by the Cartesian product of discrete parameters [41]. MERCBO similarly exploits the combinatorial graph, but with Mercer features and Thompson sampling [16]. HYBO extends the diffusion kernels to mixed continuous-discrete spaces [15]. However, these methods scale poorly with respect to the number of data points and parameters. Moreover, of the methods listed above, only HYBO supports continuous parameters without restricting them to a discrete set. HYBO enjoys a universal function approximation property, but relies on summing over all possible orders of interactions between base kernels for each parameter which results in exponential complexity with respect to the number of parameters and limits its applicability to low-dimensional problems. Moreover, the computational issues of such approaches make it difficult to apply them to multi-objective and constrained optimization. GRYFFIN [29] uses kernel density estimation, but is limited to categorical search spaces. MIVABO uses a linear combination of basis functions (e.g. pseudo Boolean features [7] for discrete parameters) with interaction terms[13]. MVRSM [6] uses ReLU-based surrogates for computational efficiency, but is limited by the expressiveness of these models.

**Optimizing acquisition functions**: As discussed previously, Garrido-Merchán and Hernández-Lobato [26] propose using continuous relaxation and discretize the inputs before evaluating the AF. However, the resulting AF after discretization is piece-wise-flat along slices of the continuous relaxation of the discrete parameters and therefore is difficult to optimize. COCABO [48] samples discrete parameters using a multi-armed bandit and optimizes the continuous parameters conditional upon the sampled discrete parameters. However, COCABO's performance degrades as number of discrete configurations increases. CASMOPOLITAN [57] uses local trust regions combined with an interleaved AF optimization strategy that alternates between local search steps on the discrete parameters and gradient ascent for the continuous parameters. Furthermore, both COCABO and CASMOPOLITAN do not inherently exploit ordinal structure.

**Probabilistic reparameterization**: PR has been considered for optimizing discrete parameters in other domains such as reinforcement learning [62] and sparse regression [65]. However, PR has not been leveraged for BO. Although the reparameterization trick used by Wilson et al. [63] is in a similar vein to PR, Wilson et al. [63] reparameterize an existing multivariate normal random variable in terms of standard normal random variables and then use sample-path gradient estimators. In contrast, our approach introduces a new probabilistic formulation using discrete probability distributions and uses likelihood-ratio-based gradient estimators since sample-path gradients cannot be computed through discrete sampling.

**Alternative methods for propagating gradients**: Alternative methods for propagating gradients through discrete structures have been considered in the deep learning community (among others).

One approach is to use approximate discrete Concrete distributions [31, 37], which admit sample-path gradients. However, samples from Concrete distributions are not discrete and approximation error can result in pathologies similar to evaluating the AF using continuous relaxation. Moreover, approximately discrete samples prohibit using surrogate models that require discrete inputs (without discretizing the samples)—e.g., GPs with Hamming distance kernels [48]. Another approach for gradient propagation in the deep learning community is to use straight-through gradient estimators (STE) [3], where the gradient of the discretization function with respect to its input is estimated using, for example, an identity function. This approach works well empirically in some cases, these estimators are not well-grounded theoretically. Nevertheless, we discuss and evaluate using STE for AF optimization in Appendix H.

## 6 Experiments

In this section, we provide an empirical evaluation of PR on a suite of synthetic problems and real world applications. For PR, we use stochastic mini-batches of $N = 128$ MC samples in our experiments and demonstrate that PR is robust with respect to the number of MC samples (and compare against analytic PR, where computationally feasible) in Appendix F. We optimize PR using Adam [34] with an initial learning rate of $\frac{1}{40}$. We show that PR Adam is generally robust to the choice of learning rate (more so than vanilla stochastic gradient ascent) in the sensitivity analysis in Figure 21 in Appendix M. We compare PR against two alternative acquisition optimization strategies: using a continuous relaxation (CONT. RELAX.) and using exact discretization with approximate gradients (EXACT ROUND) [26]. These approaches optimize the acquisition function with L-BFGS-B with exact and approximate gradients, respectively. In addition, we compare against two state-of-the-art methods for discrete/mixed BO: a modified version of CASMOPOLITAN [57] that additionally supports ordinal variables introduced in Wan et al. [58] and HYBO [15], both of which are shown to outperform the other related works discussed in Section 5. In addition, we showcase how PR is complementary to existing methods such as trust region methods [22]. We demonstrate this by using PR with a trust region for the continuous and discrete ordinal parameters and optimize PR within this trust region. In Appendix H, we provide comparison of TR methods with alterative optimizers and find that PR is the best optimizer when using TRs on 6 of the 7 benchmark problems. See Appendix C for additional discussion of PR + TR. For PR, EXACT ROUND, and PR + TR we use the sum of a product kernel and a sum kernel of a categorical kernel [48] for the categorical parameters and Matérn-5/2 kernel for all other parameters.[3] Alternative kernels over different representations of categorical parameters such as one-hot encoded vectors, latent embeddings [67], and known embeddings (e.g. using fingerprint-based reaction encodings for categorical parameters in chemical reaction optimization [51]) are evaluated in Appendix J.

CONT. RELAX., EXACT ROUND, PR, and PR + TR use expected improvement [24, 32] for single objective (constrained problems) and expected hypervolume improvement [20] for the multi-objective oil sorbent problem (where exact gradients with respect to continuous parameters are computed using auto-differentiation [8]). We report the mean for each method $\pm$ 2 standard errors across 20 replications. Performance is evaluated in terms of regret (feasible regret for constrained problems and hypervolume regret for multi-objective problems). CASMOPOLITAN and HYBO are not run on Welded Beam and Oil Sorbent as they do not support constrained and multi-objective optimization. We also leave the multi-objective extension of PR+TR to future work because it would add additional complexity [11]. For HYBO, we only run 60 BO iterations on SVM due to the large wall time (see Figure 3) and only report partial results on Cellular Network due to a singular covariance matrix error. See Appendix C for details on the experiment setup, regret metrics, benchmark problems, and methodological details. We leverage existing open source implementations of CASMOPOLITAN and HYBO (see Appendix C for links), and the implementations of all of other methods are available at `https://github.com/facebookresearch/bo_pr`.

### 6.1 Synthetic Problems

We evaluate all methods on 3 synthetic problems. **Ackley** is a 13-dimensional function with 10 binary and 3 continuous parameters (a modified version of the problem in Bliek et al. [6]). **Mixed Int F1** is a 16-dimensional variant of the F1 function from Tušar et al. [56] with 2 binary, 6 discrete ordinal parameters, and 8 continuous parameters. The discrete ordinal parameters have following

---

[3]CONT. RELAX. is incompatible with a categorical kernel, so we use a Matérn-5/2 with one-hot encoded categorical parameters.

cardinalities: 2 parameters with 3 values, 2 with 5 values, and 2 with 7 values. **Rosenbrock** is a 10-dimensional Rosenbrock function with 6 discrete ordinal parameters with 4 values each and 4 continuous parameters.

## 6.2 Real World Problems

We consider 5 real world applications including a problem with 5 black-box outcome constraints and a 3-objective problem (see Appendix D for details on constrained and multi-objective BO).

**Welded Beam** Optimizing the design of a welded steel beam is a classical engineering optimization. In this problem, the goal is to minimize manufacturing cost subject to 5 black-box constraints on structural properties of the beam (including shear stress, bending stress, and buckling load) by tuning 6 parameters: the welding configuration (binary), the metal material type (categorical with 4 options), and 4 ordinal parameters controlling the dimensions of the beam [54].

**SVM Feature Selection** This problem involves jointly performing feature selection and hyperparameter optimization for a Support Vector Machine (SVM) trained on the CTSlice UCI data set [18, 36]. The design space for this problem involves 50 binary parameters controlling whether a particular feature is included or not, and 3 continuous hyperparameters of the SVM.

**Cellular Network Optimization** In this 30-dimensional problem, the goal is to tune the tilt (ordinal with 6 values) and transmission power (continuous) for a set of 15 antennas [50] to maximize a coverage quality metric that is a function of signal power and interference [38] over a geographic region of interest. We use the simulator from Dreifuerst et al. [17].

**Direct Arylation Chemical Synthesis** Palladium-catalysed direct arylation has generated significant interest in the pharamceutical development sector[12]. In this problem, the goal is maximize yield for a direct arylation chemical reaction by tuning 3 categorical parameters corresponding to the choice of solvent, base, and ligand, as well 2 continuous parameters controlling the temperature and concentration. We fit a surrogate model to the direct arylation dataset from Shields et al. [51] in order to facilitate continuous optimization of temperature and concentration. In Appendix J, we demonstrate that PR can leverage a kernel over fingerprint-based reaction encodings computed via density functional theory (DFT) for the categorical parameters [51].

**Electrospun Oil Sorbent** Marine oil spills can cause ecological catastrophe. One avenue for mitigating environmental harm is to design and deploy absorbent materials to capture the spilled oil. In this problem, we tune 5 ordinal parameters (3 parameters with 5 values and 2 with 4 values) and 2 continuous parameters controlling the composition and manufacturing conditions for an electrospun oil sorbent material to maximize 3 competing objectives: the oil absorbing capacity, the mechanical strength, and the water contact angle [59].

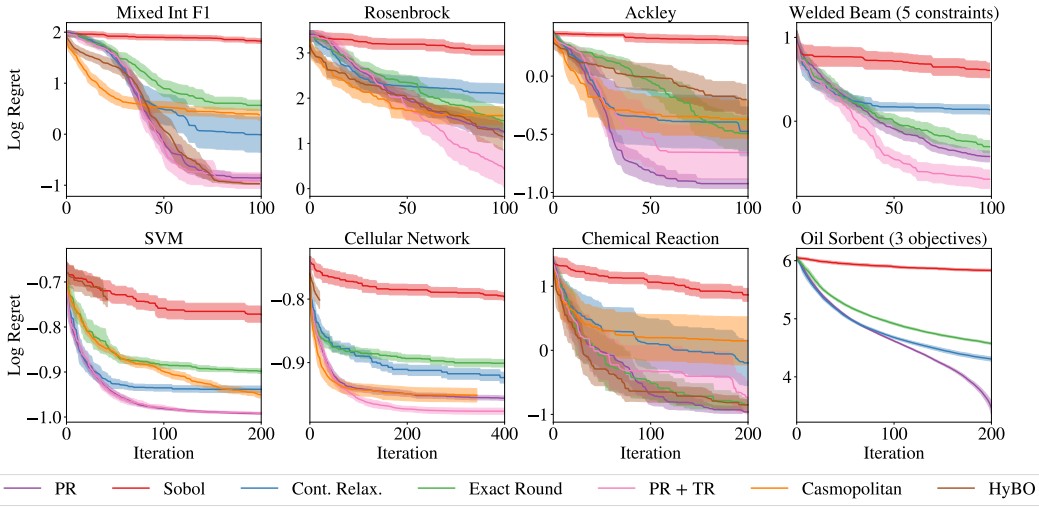

Figure 2: PR (or PR + TR) consistently outperforms alternatives with respect to log regret.

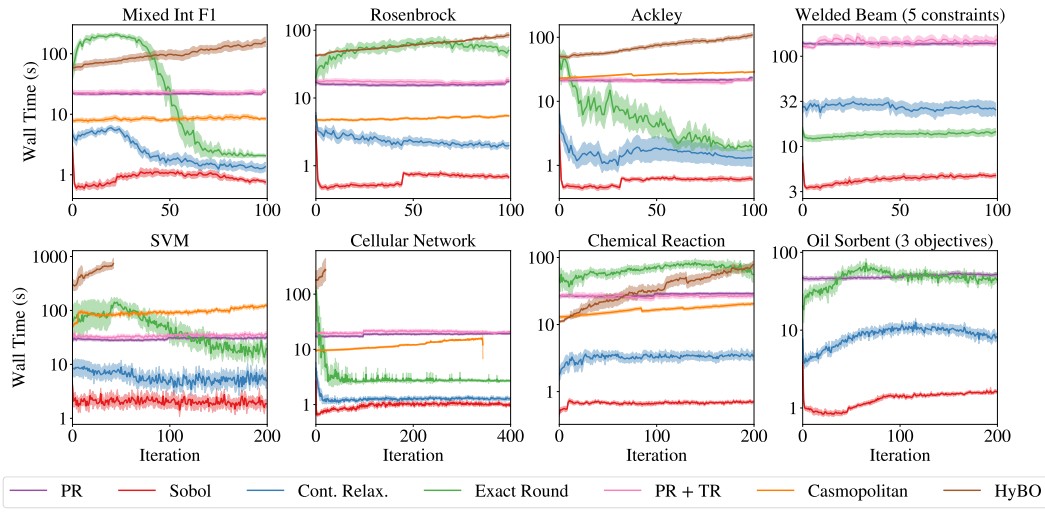

Figure 3: Wall time for candidate generation at each BO iteration in seconds. CONT. RELAX., EXACT ROUND, PR, and PR + TR are run on a single Tesla V100-SXM2-16GB GPU and other methods are run on an Intel Xeon Gold 6252N CPU.

## 6.3 Results

We find PR consistently delivers strong empirical performance as shown in Figure 2. *On all benchmark problems,* PR *(or* PR + TR*) outperforms all baseline methods (except for Mixed Int F1, where* HYBO *performs comparably).* Figure 3 shows the wall time for candidate generation over the number of BO iterations. Although PR is computationally intensive, the computation is embarrassingly parallel and therefore exploiting GPU acceleration yields competitive wall times. Importantly, PR's wall time scales well with the number of observations and design parameters, unlike HYBO which scales poorly with both. However, the complexity of PR scales additively in the number of GPs being used (e.g. outcomes being modeled), assuming they are evaluated sequentially. Hence, in multi-objective or constrained settings, PR incurs a high cost in terms of wall time. However, empirically PR achieves better optimization performance on constrained and multi-objective problems relative to CONT. RELAX. and EXACT ROUND. We note that CASMOPOLITAN does not support multi-objective BO or constrained BO, and although HYBO could be used in those settings, it would be impractically slow because 1) its wall time would scale linearly with the number of modeled outcomes (using independent GPs) and 2) its diffusion kernel is non-differentiable, which would make optimizing hypervolume-based AFs slow [8, 9].

## 7 Discussion

The performance and regret properties of BO depend critically on properly maximizing the AF. For problems with discrete features, exhaustively trying all possible combinations of discrete values quickly becomes infeasible as the number of combinations grows. Alternatives such as trying a subset of the possible combinations or resorting to continuous relaxations often leads to a failure to effectively optimize the AF which may result in sub-optimal BO performance. As an alternative, we propose using PR to better optimize the AF, and we demonstrate that PR achieves strong performance on a large number of real-world problems. Our approach is complementary to many other BO extensions, and combines seamlessly with, for example, trust region-based BO and specialized kernels for discrete parameters. One limitation of PR is that it requires computationally-demanding MC integration. However, given that the computation in PR is embarrassingly parallel, it motivates for future research on optimizing AFs on distributed hardware.

## Acknowledgements

We thank Ben Letham, James Wilson, and Michael Cohen, as well as the members of the Oxford Machine Learning Research Group, for providing insightful feedback.

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
