# Appendix to:

# Bayesian Optimization over Discrete and Mixed Spaces via Probabilistic Reparameterization

## A  Potential Societal Impacts

Our work advances Bayesian optimization, a generic class of methods for optimization of expensive, difficult-to-optimize black-box problems. With this paper in particular, we improve the performance of Bayesian optimization on problems with mixed types of inputs. Given the ubiquity of such problems in many practical applications, we believe that our method could lead to positive broader impacts by solving these problems better and more efficiently while reducing the costs incurred for solving them. Concrete and high-stake examples where our method could be potentially applied (some of which have been already demonstrated by the benchmark problems considered in the paper) include but are not limited to applications in communications, chemical synthesis, drug discovery, engineering optimization, tuning of recommender systems, and automation of machine learning systems. On the flip side, while the method proposed is ethically neutral, there is potential of misuse given that the exact objective of optimization is ultimately decided by the end users; we believe that practitioners and researchers should be aware of such possibility and aim to mitigate any potential negative impacts to the furthest extent.

## B  Theoretical Results and Proofs

### B.1  Results

Let $\mathcal{P}_{\mathcal{Z}}^{(i)} := \mathcal{P}(\mathcal{Z}^{(i)})$ denote the set of probability measures on $\mathcal{Z}^{(i)}$ for each $i = 1, ..., d_z$, and let $\mathcal{P}_{\mathcal{Z}} := \prod_{i=1}^{d_z} \mathcal{P}_{\mathcal{Z}}^{(i)}$. For any $\alpha : \mathcal{X} \times \mathcal{Z} \to \mathbb{R}$, define $\tilde{\alpha} : \mathcal{X} \times \mathcal{P} \to \mathbb{R}$ as

$$\tilde{\alpha}(\boldsymbol{x}, p) = \int_{\mathcal{Z}} \alpha(\boldsymbol{x}, \boldsymbol{z}) dp(\boldsymbol{z}) = \sum_{\boldsymbol{z} \in \mathcal{Z}} \alpha(\boldsymbol{x}, \boldsymbol{z}) p(\{\boldsymbol{z}\}). \tag{9}$$

Let $\Theta$ be a compact metric space, and consider the set of functionals $\Phi = \{\varphi \ s.t. \ \varphi : \Theta \to \mathcal{P}_{\mathcal{Z}}\}$. Let

$$\hat{\alpha}(\boldsymbol{x}, \boldsymbol{\theta}) := \tilde{\alpha}(\boldsymbol{x}, \boldsymbol{\varphi}(\boldsymbol{\theta})) = \int_{\mathcal{Z}} \alpha(\boldsymbol{x}, \boldsymbol{z}) dp_{\varphi(\boldsymbol{\theta})}(\boldsymbol{z}) = \sum_{\boldsymbol{z} \in \mathcal{Z}} \alpha(\boldsymbol{x}, \boldsymbol{z}) p_{\varphi(\boldsymbol{\theta})}(\{\boldsymbol{z}\}) \tag{10}$$

Since $\mathcal{Z}$ is finite, each element of $\varphi \in \Phi$ can be expressed as a mapping from $\Theta$ to $\mathbb{R}^{|\mathcal{Z}|}$. Namely, each $\varphi(\boldsymbol{\theta})$ corresponds to a vector with $|\mathcal{Z}|$ elements containing the probability mass for each element of $\mathcal{Z}$ under $p_\varphi(\boldsymbol{\theta})$. Thus $(\mathcal{P}_{\mathcal{Z}}, \|\cdot\|)$ is a metric space under any norm $\|\cdot\|$ on $\mathbb{R}^{|\mathcal{Z}|}$. Let $\alpha^* := \max_{(\boldsymbol{x},\boldsymbol{z}) \in (\mathcal{X} \times \mathcal{Z})} \alpha(\boldsymbol{x}, \boldsymbol{z})$ and let $\mathcal{H}^* := \arg\max_{(\boldsymbol{x},\boldsymbol{z}) \in (\mathcal{X} \times \mathcal{Z})} \alpha(\boldsymbol{x}, \boldsymbol{z})$ denote the set of maximizers of $\alpha$.

**Lemma 1.** *Suppose $\alpha$ is continuous in $\boldsymbol{x}$ for every $\boldsymbol{z} \in \mathcal{Z}$ and that $\varphi : \Theta \mapsto (\mathcal{P}_{\mathcal{Z}}, \|\cdot\|)$ is continuous with $\varphi(\Theta) = \mathcal{P}_{\mathcal{Z}}$. Let $\mathcal{J}^* := \arg\max_{(\boldsymbol{x},\boldsymbol{\theta}) \in \mathcal{X} \times \Theta} \hat{\alpha}(\boldsymbol{x}, \boldsymbol{\theta})$. Then for any $(\boldsymbol{x}^*, \boldsymbol{\theta}^*) \in \mathcal{J}^*$, it holds that $(\boldsymbol{x}^*, \boldsymbol{z}) \in \mathcal{H}^*$ for all $z \in \operatorname{supp} p_{\varphi(\boldsymbol{\theta}^*)}$.*

*Proof.* First, note that $\hat{\alpha} : \mathcal{X} \times \Theta \to \mathbb{R}$ is continuous (using that $\varphi$ is continuous and $\alpha$ is bounded). Since both $\mathcal{X}$ and $\Theta$ are compact $\hat{\alpha}$ attains its maximum, i.e., $\mathcal{J}^*$ exists. Let $(\boldsymbol{x}^*, \boldsymbol{\theta}^*) \in \mathcal{J}^*$. Clearly, there exists $\boldsymbol{z}^* \in \arg\max_{\boldsymbol{z} \in \mathcal{Z}} \alpha(\boldsymbol{x}^*, \boldsymbol{z})$ such that $\alpha(\boldsymbol{x}^*, \boldsymbol{z}^*) = \alpha^*$. Suppose there exists $\boldsymbol{z}' \in \operatorname{supp} p_{\varphi(\boldsymbol{\theta}^*)}$ such that $(\boldsymbol{x}^*, \boldsymbol{z}') \notin \mathcal{H}^*$. Then $\alpha(\boldsymbol{x}^*, \boldsymbol{z}') < \alpha^*$ and, since $\mathcal{Z}$ is finite, $p_{\varphi(\boldsymbol{\theta}^*)}(\{\boldsymbol{z}'\}) > 0$.

Consider the probability measure $p'$ given by

$$p'(\{\boldsymbol{z}\}) = \begin{cases} 0 & \text{if } \boldsymbol{z} = \boldsymbol{z}' \\ p_{\varphi(\boldsymbol{\theta}^*)}(\{\boldsymbol{z}^*\}) + p_{\varphi(\boldsymbol{\theta}^*)}(\{\boldsymbol{z}'\}) & \text{if } \boldsymbol{z} = \boldsymbol{z}^* \\ p_{\varphi(\boldsymbol{\theta}^*)}(\{\boldsymbol{z}\}) & \text{otherwise} \end{cases}$$

Then

$$\begin{aligned}
\tilde{\alpha}(\boldsymbol{x}^*, p') - \hat{\alpha}(\boldsymbol{x}^*, \boldsymbol{\theta}^*) &= \sum_{\boldsymbol{z} \in \mathcal{Z}} \alpha(\boldsymbol{x}^*, z) p'(\{\boldsymbol{z}\}) - \hat{\alpha}(\boldsymbol{x}^*, \boldsymbol{\theta}^*) \\
&= \sum_{\boldsymbol{z} \in \mathcal{Z}} \alpha(\boldsymbol{x}^*, \boldsymbol{z}) p_{\varphi(\boldsymbol{\theta}^*)}(\{\boldsymbol{z}\}) + p_{\varphi(\boldsymbol{\theta}^*)}(\{\boldsymbol{z}'\})(\alpha(\boldsymbol{x}^*, \boldsymbol{z}^*) - \alpha(\boldsymbol{x}^*, \boldsymbol{z}')) \\
&\quad - \hat{\alpha}(\boldsymbol{x}^*, \boldsymbol{\theta}^*) \\
&= p_{\varphi(\boldsymbol{\theta}^*)}(\{\boldsymbol{z}'\})(\alpha(\boldsymbol{x}^*, \boldsymbol{z}^*) - \alpha(\boldsymbol{x}^*, \boldsymbol{z}')) \\
&> 0
\end{aligned}$$

Now $p' \in \mathcal{P}_{\mathcal{Z}}$, and so $p' = \varphi(\boldsymbol{\theta}')$ for some $\boldsymbol{\theta}' \in \Theta$. But then $\hat{\alpha}(\boldsymbol{x}^*, \boldsymbol{\theta}') > \hat{\alpha}(\boldsymbol{x}^*, \boldsymbol{\theta}^*)$. This is a contradiction. $\square$

**Corollary 2.** *Suppose the optimizer of $g$ is unique, i.e., that $\mathcal{H}^* = \{(\boldsymbol{x}^*, \boldsymbol{z}^*)\}$ is a singleton. Then the optimizer of $\hat{\alpha}$ is also unique and $\mathcal{J}^* = \{(\boldsymbol{x}^*, \boldsymbol{\theta}^*)\}$, with $p_{\varphi(\boldsymbol{\theta}^*)}(\{\boldsymbol{z}^*\}) = 1$.*

**Corollary 3.** *Consider the following mappings:*

- ***Binary:*** *$\varphi : [0, 1] \to \mathcal{P}_{\{0,1\}}$ with $p_{\varphi(\theta)}(\{1\}) = \theta$ and $p_{\varphi(\theta)}(\{0\}) = 1 - \theta$.*

- ***Ordinal:*** *$\varphi : [0, C-1] \to \mathcal{P}_{\{0,1,\ldots,C\}}$ with $p_{\varphi(\theta)}(\{i\}) = (1 - |i - \theta|)\,\mathbf{1}\{|i - \theta| \leq 1\}$ for $i = 1, \ldots, C$.*

- ***Categorical:*** *$\varphi : [0, 1]^C \to \mathcal{P}_{\{0,1,\ldots,C\}}$ with $p_{\varphi(\theta)}(\{i\}) = \frac{\theta_i}{\sum_{i=1}^{C} \theta_i}$.*

*These mappings satisfy the conditions for Lemma 1. In the setting with multiple discrete parameters where the above mappings are applied in component-wise fashion for each discrete parameter, the component-wise mappings also satisfy the conditions for Lemma 1.*

Clearly, the mappings given in Corollary 3 are continuous functions of $\theta$. In the setting with multiple discrete parameters , the component-wise function is also continuous with respect to the distribution parameters for each discrete parameter. Hence, the mappings satisfy the conditions for Lemma 1.

**Lemma 2.** *If $(\boldsymbol{x}^*, \boldsymbol{z}^*) \in \mathcal{H}^* = \arg\max_{(\boldsymbol{x}, \boldsymbol{z}) \in \mathcal{X} \times \mathcal{Z}} \alpha(\boldsymbol{x}, \boldsymbol{z})$, then*

$$\alpha(\boldsymbol{x}^*, \boldsymbol{z}^*) = \max_{\boldsymbol{\theta}} \mathbb{E}_{\boldsymbol{Z} \sim p(\boldsymbol{Z}|\boldsymbol{\theta})}[\alpha(\boldsymbol{x}^*, \boldsymbol{Z})].$$

*Proof.* For any $\boldsymbol{z}^*$, let $\boldsymbol{\theta}^*$ be the parameters such that $p(\boldsymbol{z}^*|\boldsymbol{\theta}^*) = 1$ (i.e. a point mass on $\boldsymbol{z}^*$). From Equation (2),

$$\mathbb{E}_{\boldsymbol{Z} \sim p(\boldsymbol{Z}|\boldsymbol{\theta}^*)}[\alpha(\boldsymbol{x}^*, \boldsymbol{Z})] = \sum_{\boldsymbol{z} \in \mathcal{Z}} \alpha(\boldsymbol{x}^*, \boldsymbol{z}) p(\boldsymbol{z}|\boldsymbol{\theta}^*) = \alpha(\boldsymbol{x}^*, \boldsymbol{z}^*).$$

Claim: $\mathbb{E}_{\boldsymbol{Z} \sim p(\boldsymbol{Z}|\boldsymbol{\theta}^*)}[\alpha(\boldsymbol{x}^*, \boldsymbol{Z})] = \max_{\boldsymbol{\theta}} \mathbb{E}_{\boldsymbol{Z} \sim p(\boldsymbol{Z}|\boldsymbol{\theta})}[\alpha(\boldsymbol{x}^*, \boldsymbol{Z})]$.

Suppose there exists $\boldsymbol{\theta}'$ such that $\mathbb{E}_{\boldsymbol{Z} \sim p(\boldsymbol{Z}|\boldsymbol{\theta}')}[\alpha(\boldsymbol{x}^*, \boldsymbol{Z})] > \mathbb{E}_{\boldsymbol{Z} \sim p(\boldsymbol{Z}|\boldsymbol{\theta}^*)}[\alpha(\boldsymbol{x}^*, \boldsymbol{Z})]$. Since $(\boldsymbol{x}^*, \boldsymbol{z}^*) \in \mathcal{H}^*$, $\alpha(\boldsymbol{x}^*, \boldsymbol{z}^*) = \max_{(\boldsymbol{x}, \boldsymbol{z}) \in \mathcal{X} \times \mathcal{Z}} \alpha(\boldsymbol{x}, \boldsymbol{z})$. Hence, there is no convex combination of values of $\alpha$ that is greater than $\alpha(\boldsymbol{x}^*, \boldsymbol{z}^*)$. This is a contradiction. $\square$

**Theorem 1** (Consistent Maximizers). *Suppose that $\alpha$ is continuous in $\boldsymbol{x}$ for every $\boldsymbol{z} \in \mathcal{Z}$. Let $\mathcal{H}^*$ be the maximizers of $\alpha(\boldsymbol{x}, \boldsymbol{z})$: $\mathcal{H}^* = \{(\boldsymbol{x}, \boldsymbol{z}) \in \arg\max_{(\boldsymbol{x}, \boldsymbol{z}) \in \mathcal{X} \times \mathcal{Z}} \alpha(\boldsymbol{x}, \boldsymbol{z})\}$. Let $\mathcal{J}^* \subseteq \mathcal{X} \times \Theta$ be the maximizers of $\mathbb{E}_{\boldsymbol{Z} \sim p(\boldsymbol{Z}|\boldsymbol{\theta})}[\alpha(\boldsymbol{x}, \boldsymbol{Z})]$: $\mathcal{J}^* = \{(\boldsymbol{x}, \boldsymbol{\theta}) \in \arg\max_{(\boldsymbol{x}, \boldsymbol{\theta}) \in \mathcal{X} \times \Theta} \mathbb{E}_{\boldsymbol{Z} \sim p(\boldsymbol{Z}|\boldsymbol{\theta})}[\alpha(\boldsymbol{x}, \boldsymbol{Z})]\}$, where $\Theta$ is the domain of $\boldsymbol{\theta}$. Let $\hat{\mathcal{H}}^* \subseteq \mathcal{X} \times \mathcal{Z}$ be defined as: $\hat{\mathcal{H}}^* = \{(\boldsymbol{x}, \tilde{\boldsymbol{z}}) : (\boldsymbol{x}, \boldsymbol{\theta}) \in \mathcal{J}^*, \tilde{\boldsymbol{z}} \sim p(\boldsymbol{Z}|\boldsymbol{\theta})\}$. Then, $\hat{\mathcal{H}}^* = \mathcal{H}^*$.*

*Proof.* From Lemma 1, we have that for any $(\boldsymbol{x}^*, \boldsymbol{\theta}^*) \in \mathcal{J}^*$, it holds that $(\boldsymbol{x}^*, \boldsymbol{z}) \in \mathcal{H}^*$ for all $z \in \operatorname{supp} p_{\varphi(\boldsymbol{\theta}^*)}$. Hence, $\hat{\mathcal{H}}^* \subseteq \mathcal{H}^*$.

Now, let $(\boldsymbol{x}^*, \boldsymbol{z}^*) \in \mathcal{H}^*$. Let $\boldsymbol{\theta}^* \in \Theta$ such that $p(\boldsymbol{z}^*|\boldsymbol{\theta}^*) = 1$. From the proof of Lemma 2, we have that $\mathbb{E}_{\boldsymbol{Z} \sim p(\boldsymbol{Z}|\boldsymbol{\theta}^*)}[\alpha(\boldsymbol{x}^*, \boldsymbol{Z})] = \alpha(\boldsymbol{x}^*, \boldsymbol{z}^*)$. As in the proof of Lemma 2, there is no convex combination of values of $\alpha$ greater than $\alpha(\boldsymbol{x}^*, \boldsymbol{z}^*)$. So $\mathbb{E}_{\boldsymbol{Z} \sim p(\boldsymbol{Z}|\boldsymbol{\theta}^*)}[\alpha(\boldsymbol{x}^*, \boldsymbol{Z})] = \max_{(\boldsymbol{x}, \boldsymbol{\theta}) \in \mathcal{X} \times \Theta} \mathbb{E}_{\boldsymbol{Z} \sim p(\boldsymbol{Z}|\boldsymbol{\theta})}[\alpha(\boldsymbol{x}, \boldsymbol{Z})]$, and therefore, $\boldsymbol{x}^*, \boldsymbol{\theta}^* \in \mathcal{J}^*$. Hence $(\boldsymbol{x}^*, \boldsymbol{z}^*) \in \hat{\mathcal{H}}^*$. So $\mathcal{H}^* \subseteq \hat{\mathcal{H}}^*$, and hence, $\hat{\mathcal{H}}^* = \mathcal{H}^*$. $\qquad\square$

**Lemma 3.** *Suppose that $\alpha : (\boldsymbol{x}, \boldsymbol{z}) \mapsto \mathbb{R}$ is differentiable with respect to $\boldsymbol{x}$ for all $\boldsymbol{z} \in \mathcal{Z}$, and that the mapping $\varphi : \boldsymbol{\theta} \mapsto \mathcal{P}_{\mathcal{Z}}$ is such that $p_{\varphi(\boldsymbol{\theta})}(\{\boldsymbol{z}\})$ is differentiable with respect to $\boldsymbol{\theta}$ for all $z \in \mathcal{Z}$. Then the probabilistic objective $\mathbb{E}_{\boldsymbol{Z} \sim p(\boldsymbol{Z}|\boldsymbol{\theta})}[\alpha(\boldsymbol{x}, \boldsymbol{Z})]$ is differentiable with respect to $(\boldsymbol{x}, \boldsymbol{\theta})$.*

*Proof.* For any $\boldsymbol{z} \in \mathcal{Z}$, the function $p(\boldsymbol{z}, \boldsymbol{\theta})\alpha(\boldsymbol{x}, \boldsymbol{z}) = p_{\varphi(\boldsymbol{\theta})}(\{\boldsymbol{z}\})\alpha(\boldsymbol{x}, \boldsymbol{z})$ is the product of two differentiable functions, hence differentiable. Therefore the probabilistic objective is a (finite) linear combination of differentiable functions, hence differentiable. $\qquad\square$

**Theorem 2** (Convergence Guarantee). *Let $\alpha : \mathcal{X} \times \mathcal{Z} \to \mathbb{R}$ be differentiable in $\boldsymbol{x}$ for every $\boldsymbol{z} \in \mathcal{Z}$. Let $(\hat{\boldsymbol{x}}_{t,m}, \hat{\boldsymbol{\theta}}_{t,m})$ be the best solution after running stochastic gradient ascent for $t$ time steps on the probabilistic objective $\mathbb{E}_{\boldsymbol{Z} \sim p(\boldsymbol{Z}|\boldsymbol{\theta})}[\alpha(\boldsymbol{x}, \boldsymbol{Z})]$ from $m$ starting points with its unbiased MC estimators proposed above. Let $\{a_t\}_{t=1}^{\infty}$ be a sequence of positive step sizes such that $0 < \sum_{t=1}^{\infty} a_t^2 = A < \infty$ and $\sum_{t=1}^{\infty} a_t = \infty$, where $a_t$ is the step size used in stochastic gradient ascent at time step $t$. Let $\hat{z}_{t,m} \sim p(\boldsymbol{Z}|\hat{\boldsymbol{\theta}}_{t,m})$. Then as $t \to \infty$, $m \to \infty$, and $\tau \to 0$, $(\hat{\boldsymbol{x}}_{t,m}, \hat{\boldsymbol{z}}_{t,m}) \to (\boldsymbol{x}^*, \boldsymbol{z}^*) \in \arg\max_{(\boldsymbol{x}, \boldsymbol{z}) \in \mathcal{X} \times \mathcal{Z}} \alpha(\boldsymbol{x}, \boldsymbol{z})$ in probability.*

*Proof.* The binary and categorical mappings in Corollary 3 are differentiable in $\theta$ (the ordinal mapping is differentiable almost everywhere[4]). Since the acquisition function $\alpha : \mathcal{X} \times \mathcal{Z} \to \mathbb{R}$ is differentiable in $\boldsymbol{x}$ for every $\boldsymbol{z} \in \mathcal{Z}$, this means that the PO is differentiable. Using the prescribed sequence of step sizes, optimizing the PO using stochastic gradient ascent will converge almost surely to a local maximum after a sufficient number of steps [47]. As we increase the number of randomly distributed starting points, the probability of not finding the global maximum of the PO will converge to zero [60]. From Theorem 1, the PO and the AF have the same set of maximizers. Hence, convergence in probability to a global maximizer of the PO means convergence in probability to a global maximizer of the AF. $\qquad\square$

## C   Experiment Details

For each BO optimization replicate, we use $N_{\text{init}} = \min(20, 2 * d_{\text{eff}})$ points from a scrambled Sobol sequence, where $d_{\text{eff}}$ is the "effective dimensionality" after one-hot encoding categorical parameters. Unless otherwise noted, all experiments use 20 replications and confidence intervals represent 2 standard errors of the mean. The same initial points are used for all methods for that replicate and different initial points are used for each replicate. For each method we report the $\log_{10}$ regret. Since the optimal value is unknown for many problems, we set the optimal value to be $f^* + 0.1$ where $f^*$ is the best observed value across all methods and all replications. For constrained optimization $f^*$ is the best feasible observed value and for multi-objective optimization $f^*$ is the maximum hypervolume across all methods and replications. In total, the experiments in the main text (excluding HYBO and CASMOPOLITAN) ran for an equivalent of 2,009.82 hours on a single Tesla V100-SXM2-16GB GPU. The baseline experiments (HYBO and CASMOPOLITAN) ran for an equivalent of 745.10 hours on a single Intel Xeon Gold 6252N CPU.

---

[4]Technically, the arguments presented here do not prove convergence under the ordinal mapping, but we have found this to work well and reliably in practice. Alternatively, ordinal parameters could also just be treated as categorical ones in which case the convergence results hold. In practice, however, this introduces additional optimization variables that make the problem unnecessarily hard by removing the ordered structure from the problem.

## C.1 Additional Problem Details

In this section, we describe the details of each synthetic problem considered in the experiments (the details of the remaining real-world problems are already described in Section 6.2).

**Ackley.** We use an adapted version of the 13-dimensional Ackley function modified from Bliek et al. [6]. The function is given by:

$$f(\mathbf{x}) = -a \exp\left(-b\sqrt{\frac{1}{d}\sum_{i=1}^{d} x_i^2}\right) - \exp\left(\frac{1}{d}\sum_{i=1}^{d}\cos(cx_i)\right) + a + \exp(1), \tag{11}$$

where in this case $a = 20, n = 0.2, c = 2\pi$ and $d = 13$ and $\mathbf{x} \in [-1,1]^{13}$. We discretize the first 10 dimensions to be binary with the choice $\{-1,1\}$, and the final 3 dimensions are unmodified with the original bounds.

**Mixed Int F1.** Mixed Int F1 is a partially discretized version of the 16-dimensional Sphere optimization problem [28], given by:

$$f(\mathbf{x}) = \sum_{i=1}^{d} (x_i - x_{\text{opt},i})^2 + f_{\text{opt}}, \tag{12}$$

where $f_{\text{opt}}$ is sampled from a Cauchy distribution with median = 0 and roughly 50% of the values between $-100$ and $100$. The sampled $f_{\text{opt}}$ is then clamped to be between $[-1000, 1000]$ and rounded to the nearest integer. $\mathbf{x}_{\text{opt}}$ is sampled uniformly in $[-4,4]^d$, and in this case $d = 16$. We discretize the first 8 dimensions as follows: the first 2 dimensions are binary with 2 choices $\{-5,5\}$; the next 2 dimensions are ordinal with 3 choices $\{-5,0,5\}$; the next 2 dimensions are ordinal with 5 choices $\{-5,-2.5,0,2.5,5\}$; the final 2 dimensions are ordinal with 7 choices $\{-5,-\frac{10}{3},-\frac{5}{3},0,\frac{5}{3},\frac{10}{3},5\}$. The remaining 8 dimensions are continuous with bounds $[-5,5]^8$.

**Rosenbrock.** We use an adapted version of the Rosenbrock function, given by:

$$f(\mathbf{x}) = \left(\sum_{i=1}^{d-1}\left(100(x_{i+1} - x_i^2)^2 + (x_i - 1)^2\right)\right), \tag{13}$$

where in this case $d = 10$. The first 6 dimensions are discretized to be ordinal variables, with 4 possible values each $x_i \in \{-5,0,5,10\} \forall i \in [1,6]$. The final 4 dimensions are continuous with bounds $[-5,10]^4$.

**Chemical Reaction (Direct Arylation Chemical Synthesis).** For this problem, we fit a GP surrogate (with the same kernel used by the BO methods) to the dataset from Shields et al. [51] (available at `https://github.com/b-shields/edbo/tree/master/experiments/data/direct_arylation` under the MIT license) in order to facilitate continuous optimization of temperature and concentration. The surrogate is included with our source code.

**Oil Sorbent.** We set the reference point for this problem to be $[-125.3865, -57.8292, 43.2665]$, which we choose using a commonly used heuristic to scale the nadir point (component-wise worst objective values across the Pareto frontier) [61].

## C.2 Method details

**PR, CONT. RELAX., EXACT ROUND, PR + TR, and EXACT ROUND + STE**. We implemented all of these methods using BoTorch [1], which is available under the MIT license at `https://github.com/pytorch/botorch`. PR and PR + TR use stochastic minibatches of 128 samples and the probabilistic objectives are optimized via Adam using a learning rate of $\frac{1}{40}$. The AFs of CONT. RELAX., EXACT ROUND, EXACT ROUND + STE are deterministic and are optimized via L-BFGS-B—EXACT ROUND approximates gradients via finite differences [26]. All methods use 20 random restarts and are run for a maximum of 200 iterations. We follow the default initialization heuristic in BoTorch [1], which initializes the optimizer by evaluating the acquisition function at a

large number of starting points (here, 1024, chosen from a scrambled Sobol sequence), and selecting (20) points using Boltzmann sampling [19] of the 1024 initial points, according to their acquisition function utilities.

**Combining PR with trust regions**: When combining PR with the trust regions used in TURBO we only use a trust region over the continuous parameters and discrete ordinals with at least 3 values. While methods like CASMOPOLITAN uses a Hamming distance for the trust regions over the categorical parameters, we choose not to do so as there is no natural way of efficiently optimizing PR using gradient-based methods. Finally, we do not use a trust region over the Boolean parameters as the trust region will quickly shrink to only include one possible value. We use the same hyperparameters as TURBO [22] for unconstrained problems and SCBO [21] in the presence of outcome constraints, including default trust region update settings.

**Casmopolitan**: We use the implementation of CASMOPOLITAN—which is available at `https://github.com/xingchenwan/Casmopolitan` under the MIT licence—but modify it where appropriate to additionally handle the ordinal variables. Specifically, the ordinal variables are treated as continuous when computing the kernel. However, during interleaved search, ordinal variables are searched via local search similar to the categorical variables. We use a set of CASMOPOLITAN hyperparameters (i.e. success/failure sensitivity, initial trust region sizes and expansion factor) recommended by the authors. We use the same implementation of interleaved search for the acquisition optimization comparisons.

**HyBO**: We use the official implementation of HYBO at `https://github.com/aryandeshwal/HyBO`, which is licensed by the University of Amsterdam. We use the default hyperparameters recommended by the authors in all the experiments, and we use the full HYBO method with marginalization treatment of the hyperparameters as it has been shown to perform stronger empirically [14].

## C.3 Gaussian process regression

When there are no categorical variables we use $k_{\text{ordinal}}$ which is a product of an isotropic Matern-5/2 kernel for the binary parameters and a Matern-5/2 kernel with ARD for the remaining ordinal parameters. In the presence of categorical parameters, this kernel is combined with a categorical kernel [48] $k_{\text{cat}}$ as $k_{\text{cat}} \times k_{\text{ordinal}} + k_{\text{cat}} + k_{\text{ordinal}}$. We use a constant mean function. The GP hyperparameters are fitted using L-BFGS-B by optimizing the log-marginal likelihood. The ranges for the ordinal parameters are rescaled to $[0, 1]$ and the outcomes are standardized before fitting the GP.

## C.4 Variance Reduction via Control Variates

As discussed in Section 4.2, we use moving average baseline for variance reduction. Specifically, the baseline is an exponential moving average with a multiplier of 0.7, where each element is the mean acquisition value across the $N$ MC samples obtained while evaluating the probabilistic objective.

## C.5 Deterministic Optimization via Sample Average Approximation

Although multi-start stochastic ascent is provably convergent, an alternative optimization approach is to use common random numbers (i.e. a fixed set of base samples) to reduce variance when comparing a stochastic function at different inputs by using the same random numbers. The method of common random numbers leads to biased deterministic estimators that are lower-variance than their stochastic counterparts where random numbers are re-sampled at each step. Such techniques have been used in the context of BO in settings such as efficiently optimizing MC acquisition functions [1] and for optimizing risk measures of acquisition functions under random inputs [10].

Sampling a fixed set of points $\tilde{z}_1, ..., \tilde{z}_N \sim p(\boldsymbol{Z}|\boldsymbol{\theta})$ would be a poor choice because $p(\boldsymbol{Z}|\boldsymbol{\theta})$ can vary widely during AF optimization as $\boldsymbol{\theta}$ changes. Therefore, instead sample from $p(\boldsymbol{Z}|\boldsymbol{\theta})$ using reparameterizations provided in Table 4. Specifically, we reparameterize $\boldsymbol{Z}$ as a deterministic function $h(\cdot, \cdot)$ that operates component-wise on $\boldsymbol{\theta}$ and the random variable $\boldsymbol{U} = (u^{(1)}, ..., u^{(d_z)}), u^{(i)} \sim$ Uniform$(0, 1)$: $\boldsymbol{Z} = h(\boldsymbol{\theta}, \boldsymbol{U})$. That is, each random variable $Z^{(j)}$, where $j = 1, ..., d_z$ has a corresponding independent base random variable $U^{(j)}$ such that $Z^{(j)} = h(\theta^{(j)}, U^{(j)})$. Using a fixed a set of base samples $\{\tilde{\boldsymbol{u}}_i\}_{i=1}^{N}$, the samples of $\boldsymbol{Z}$ can be be computed as $\boldsymbol{z}_i = h(\boldsymbol{\theta}, \tilde{\boldsymbol{u}}_i)$. We note that

even with fixed base samples, the samples $\{z_i\}_{i=1}^N$ depends on $\boldsymbol{\theta}$, and hence, by using common *base* uniform samples, we obtain a deterministic estimator where the values of the samples $\tilde{z}_1, ..., \tilde{z}_N$ can still vary with $\boldsymbol{\theta}$. Under this reparameterization, our probabilistic objective can be written as

$$\mathbb{E}_{\boldsymbol{Z} \sim p(\boldsymbol{Z}|\boldsymbol{\theta})}[\alpha(\boldsymbol{x}, \boldsymbol{Z})] = \mathbb{E}_{\boldsymbol{U} \sim p(\boldsymbol{U})}[\alpha(\boldsymbol{x}, h(\boldsymbol{\theta}, \boldsymbol{U}))], \tag{14}$$

where under the reparameterizations in Table 4, $U$ is a uniform random variable across the $d_z$-dimensional unit cube—$P(\boldsymbol{U}) = \text{Uniform}(0,1)_z^d$. Under this reparameterization we can define our sample average approximation estimator of the probabilistic objective as

$$\mathbb{E}_{\boldsymbol{Z} \sim p(\boldsymbol{Z}|\boldsymbol{\theta})}[\alpha(\boldsymbol{x}, \boldsymbol{Z})] \approx \frac{1}{N} \sum_{i=1}^N \alpha(\boldsymbol{x}, h(\boldsymbol{\theta}, \tilde{\boldsymbol{u}}_i)). \tag{15}$$

Our sample average approximation estimator of the gradient of the probabilical objective with respect to $\boldsymbol{\theta}$ is given by

$$\nabla_{\boldsymbol{\theta}} \mathbb{E}_{\boldsymbol{Z} \sim p(\boldsymbol{Z}|\boldsymbol{\theta})}[\alpha(\boldsymbol{x}, \boldsymbol{Z})] \approx \frac{1}{N} \sum_{i=1}^N \alpha(\boldsymbol{x}, h(\boldsymbol{\theta}, \tilde{\boldsymbol{u}}_i)) \nabla_{\boldsymbol{\theta}} \log p(h(\boldsymbol{\theta}, \tilde{\boldsymbol{u}}_i)|\boldsymbol{\theta}). \tag{16}$$

Sample average approximation estimators are deterministic and biased conditional on the selection of base samples. However, the reparameterizations in Table 4 create discontinuities in the PO, and the number of discontinuities increases with the number of MC samples. Nevertheless, we find that optimizing the PO using L-BFGS-B delivers strong performance on the benchmark problems and we compare against stochastic optimization in Figures 21 and 20. As in the stochastic case, we reduce the variance further by leveraging quasi-MC sampling [42] instead of i.i.d. sampling.

Table 4: Discrete random variables and their reparameterizations in terms of a Uniform random variable $U \sim \text{Uniform}(0,1)$ and $\theta$ via a deterministic function $h(\cdot, \cdot)$.

| TYPE | RANDOM VARIABLE | REPARAMETERIZATION ($Z = h(\theta, U)$) |
|---|---|---|
| BINARY | $Z \sim \text{BERNOULLI}(\theta)$ | $h(\theta, U) = \mathbb{1}(U < \theta)$ |
| ORDINAL | $Z = \lfloor \theta \rfloor + B,$ | $h(\theta, U) = \lfloor \theta \rfloor + \mathbb{1}(U < \theta - \lfloor \theta \rfloor)$ |
| | $B \sim \text{BERNOULLI}(\theta - \lfloor \theta \rfloor)$ | |
| CATEGORICAL | $Z \sim \text{CATEGORICAL}(\boldsymbol{\theta})$ | $h(\theta, U) = \min(\arg\max_{i=0}^{C-1} \mathbb{1}(U < \sum_c^i \theta^{(c)}))$ |

## D  Constrained and Multi-Objective Bayesian Optimization

In many practical problems, the black-box objective must be maximized subject to $V > 0$ black-box outcome constraints $f_c^{(v)}(\boldsymbol{x}, \boldsymbol{z}) \geq 0$ for $v = 1, ..., V$. See Gardner et al. [24] for a more in depth review of black-box optimization with black-box constraints and BO techniques for this class of problems.

In the multi-objective setting, the goal is to maximize (without loss of generality) a set of $M$ objectives $f^{(1)}, ..., f^{(M)}$. Typically there is no single best solution, and hence the goal is to learn the Pareto frontier (i.e. the set of optimal trade-offs between objectives). In the multi-objective setting, the hypervolume indicator is a common metric for evaluating the quality of a Pareto frontier. See [20] for a review of multi-objective optimization.

## E  Comparison with Enumeration

When computationally feasible, the gold standard for acquisition optimization over discrete and mixed search spaces is to enumerate the discrete options and optimize any continuous parameters for each discrete configuration (or simply evaluated each discrete configuration for fully discrete spaces). In Figures 4 and 5 we compare PR (optimized with Adam using stochastic mini-batches of 128 MC samples) and analytic PR (optimized with L-BFGS-B) against enumeration and show that PR achieves log regret performance that is comparable to the gold standard of enumeration and does so in less wall time.

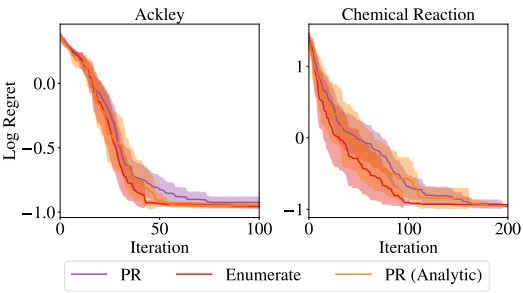

Figure 4: A comparison with an enumeration (gold standard) with respect to log regret.

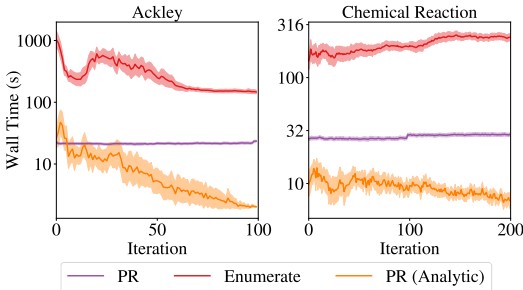

Figure 5: A comparison with enumeration with respect to wall time.

## F Analysis of MC sampling in Probabilistic Reparameterization

The main text considers 1024 MC for PR. We consider 128, 256, and 512 samples, in addition to the default of 1024. For problems with discrete spaces that are enumerable, we also consider analytic PR. We do not find statistically significant differences between the final regret of any of these configurations (Figure 6). Run time is linear with respect to MC samples, and so substantial compute savings are possible when fewer MC samples are used (Figure 7). We observe comparable performance between PR with 1,024 MC samples and as few as 128 MC samples. With 64 or fewer MC samples, we observe performance degradation with respect to log regret in Figure 8, although wall time is considerably faster for fewer 64 or less MC samples as shown in Figure 9.

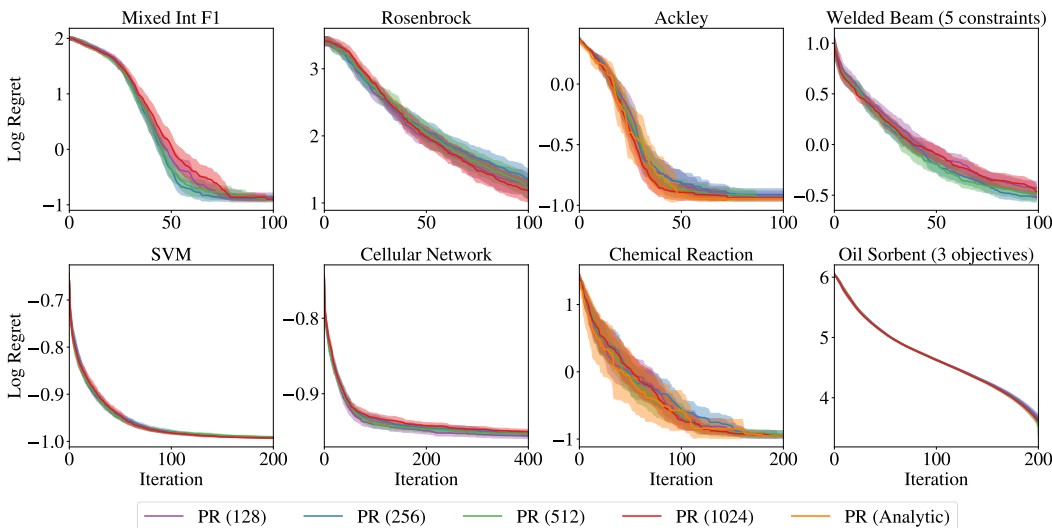

Figure 6: A sensitivity analysis of the optimization performance of PR with respect to the number of MC samples. We find that PR is robust to the number of MC samples, and that the performance of MC PR matches that of analytic PR.

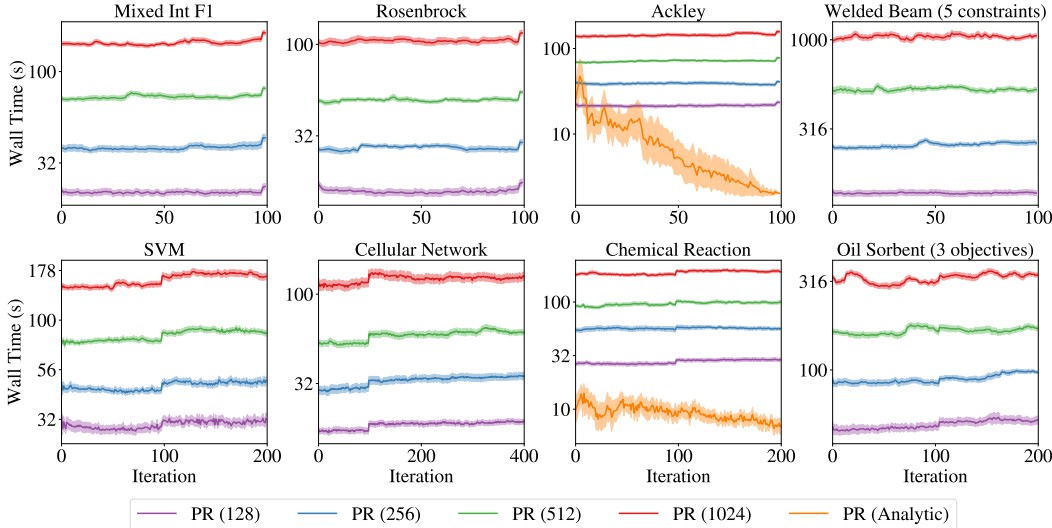

Figure 7: A sensitivity analysis of the wall time of PR with respect to the number of MC samples. We observe that wall time scales linearly with the number of MC samples, which is expected since we compute PR in $\frac{N}{32}$ chunks to avoid overflowing GPU memory.

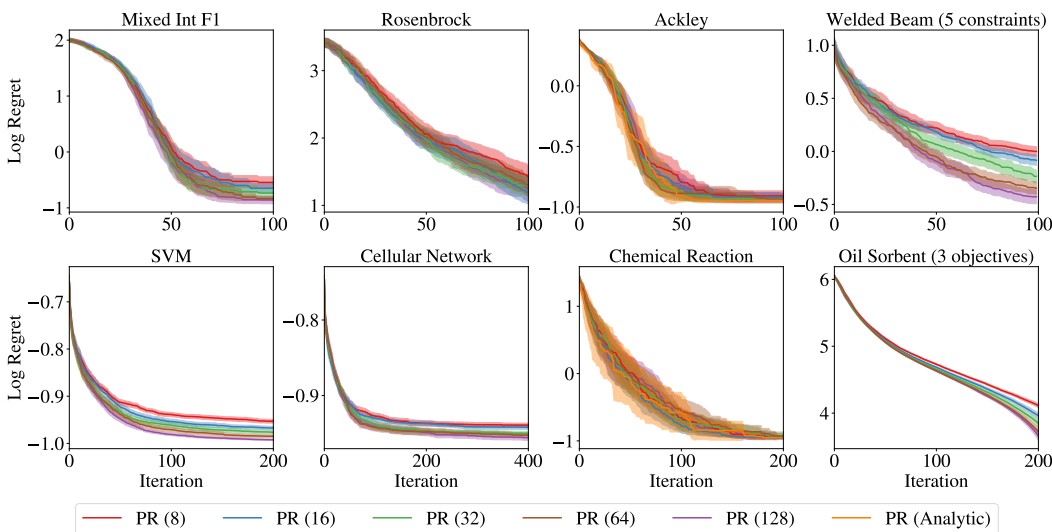

Figure 8: A sensitivity analysis of the optimization performance of PR with respect to a small number of MC samples (with samples between 8 and 64). Performance degrades slightly when few samples are used.

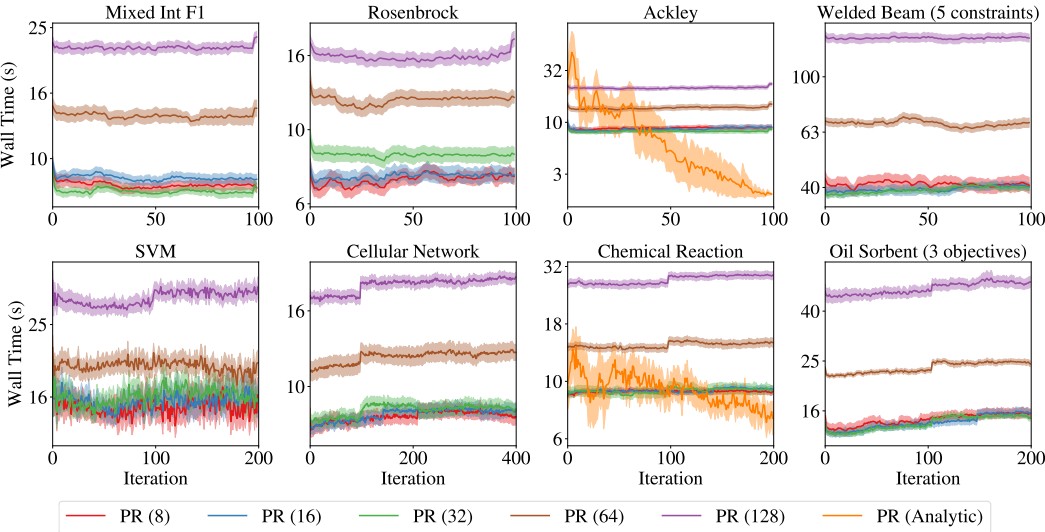

Figure 9: A sensitivity analysis of the wall time of PR with respect to the number of MC samples (with samples between 8 and 64). We observe that wall time scales linearly with the number of MC samples, which is expected since we compute PR in $\frac{N}{32}$ chunks to avoid overflowing GPU memory.

### F.0.1 Evaluation of Approximation Error in MC Sampling

We examine the MC approximation error relative to analytic PR on the chemical reaction and ackley problems. The results in Figure 10 show the mean absolute percentage error (MAPE)

$$\frac{100}{|X_{\text{discrete}}|} \cdot \sum_{\boldsymbol{x} \in X_{\text{discrete}}, \boldsymbol{\theta} \in \Theta_{\text{discrete}}} \frac{\mathbb{E}_{\boldsymbol{Z} \sim p(\boldsymbol{Z}|\boldsymbol{\theta})} \alpha(\boldsymbol{x}, \boldsymbol{Z}) - \frac{1}{N} \sum_{i=1}^{N} \alpha(\boldsymbol{x}, \tilde{\boldsymbol{z}}_i)}{\max_{\boldsymbol{x} \in X_{\text{discrete}}, \boldsymbol{\theta} \in \Theta_{\text{discrete}}} \mathbb{E}_{\boldsymbol{Z} \sim p(\boldsymbol{Z}|\boldsymbol{\theta})} \alpha(\boldsymbol{x}, \boldsymbol{Z})}$$

evaluated over a random set of $|X_{\text{discrete}}| = |\Theta_{\text{discrete}}| 10,000$ points from $\mathcal{X} \times \times$ (the sampled sets are denoted $X_{\text{discrete}}, \Theta_{\text{discrete}}$). We observe a rapid reduction in MAPE as we increase the number of samples. With 1024 samples, MAPE is 0.055% (+/- 0.0002 %) over 20 replications (different MC samples in PR) on the chemical reaction problem and MAPE is 0.018% (+/- 0.0003 %) on the ackley problem.

With 128 samples, MAPE is 0.282% (+/- 0.0029 %) over 20 replications (different MC samples in PR) on the chemical reaction problem and MAPE is 0.052% (+/- 0.0021 %) on the ackley problem.

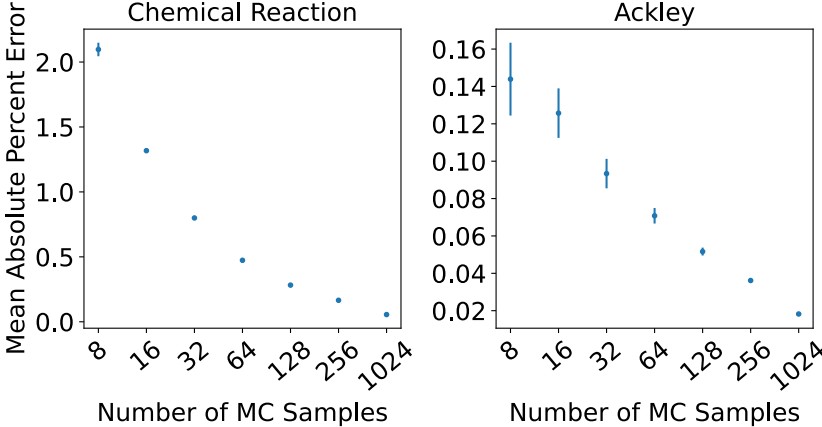

Figure 10: An evaluation of the mean absolute percentage error for the MC estimator of PR (relative to analytic PR).

## G   Effect of $\tau$ in Transformation

Throughout the main text, we use $\tau = 0.1$, which we selected based on the observation that it provides a reasonable balance between retaining non-zero gradients of $g(\phi)$ with respect to $\phi$ and allowing $\theta$ to become close to 0 or 1 as shown in Figure 11.

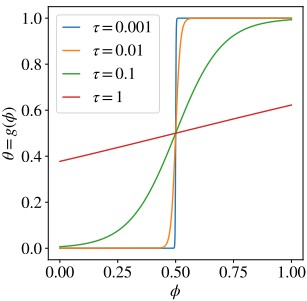

Figure 11: A comparison of the reparameterization of $\theta$ under various choices of $\tau$. We observe that $\tau = 0.1$ provides a reasonable balance between retaining non-zero gradients of $g(\phi)$ with respect to $\phi$ and allowing $\theta$ to become close to 0 or 1.

As $\tau \to 0$, the $\theta$ can take more extreme values, but the gradient of the transformation with respect to $\phi$ also moves closer to zero. For larger values of $\tau$, the gradient of the transformation with respect

to $\phi$ is larger, but $\theta$ has a more limited domain with less extreme values. We find that $\tau = 0.1$ is a robust setting across all experiments.

# H    Alternative methods

## H.1    Straight-through gradient estimators

An alternative approach to using approximating the gradients under exact rounding using finite differences is to approximate the gradients using straight-through gradient estimation (STE) [3]. The idea of STE is to approximate the gradient of a function with the identity function. In our setting, the gradient of the discretization function with respect to its input is estimated using an identity function. Using this estimator enables gradient-based AF optimization, even though the true gradient of the discretization function is zero everywhere that it is defined. Although STEs have been shown to work well empirically, these estimators are not well-grounded theoretically. Their robustness and potential pitfalls in the context of AF optimization have not been well studied. Below, we evaluate the aforementioned EXACT ROUND + STE approach and show that it offers competitive optimization performance (Figure 12) with fast wall times (Figure 13), but does not quite match the optimization performance of PR on several benchmark problems.

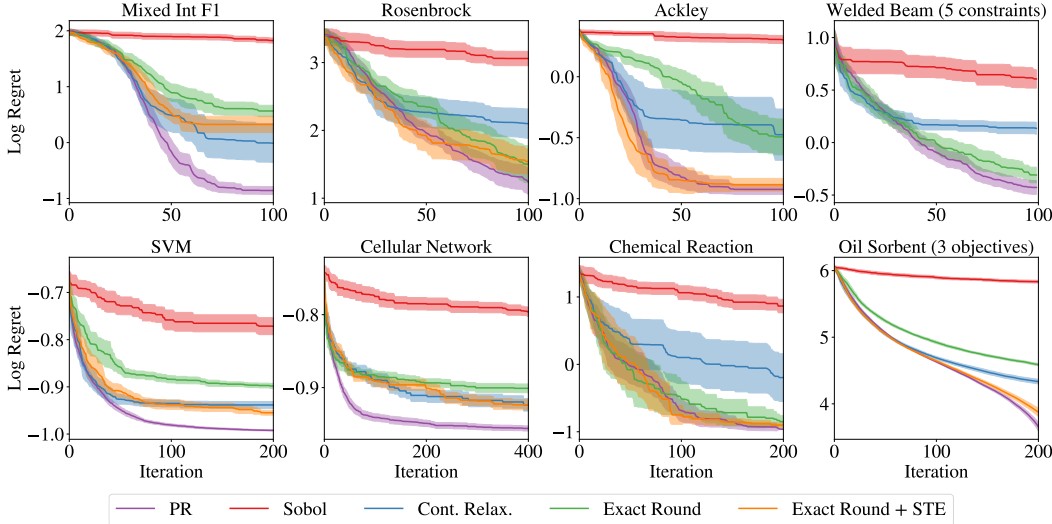

Figure 12: A comparison of exact rounding with straight-through gradient estimators versus other acquisition optimization strategies. Log regret on each problem. We report log hypervolume regret for Oil Sorbent and report the log regret of the best feasible objective for Welded beam.

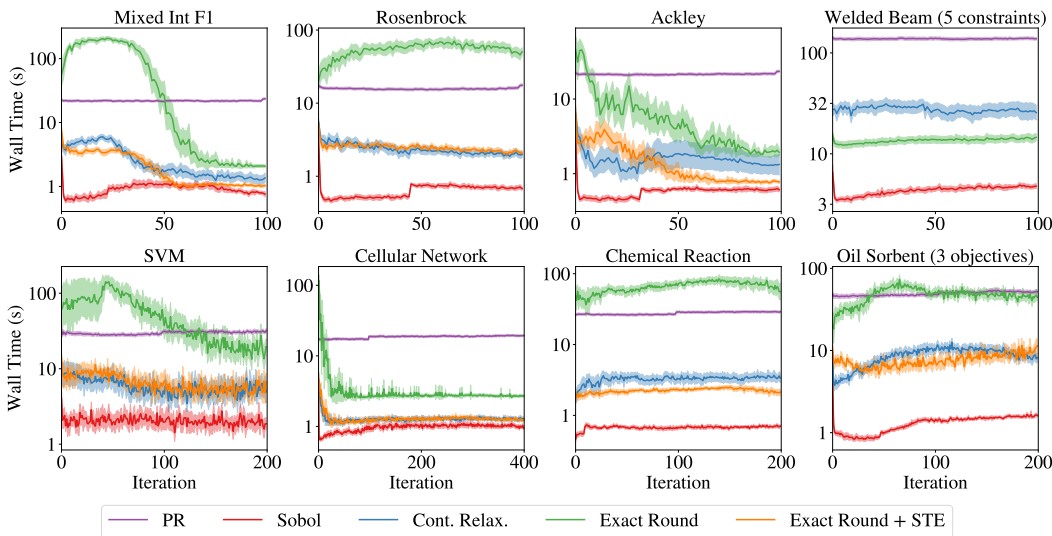

Figure 13: A comparison of wall times of exact rounding with straight-through gradient estimators versus other acquisition optimization strategies.

## H.2 TR methods with alternative optimizers

In this section, we consider alternative methods to PR for optimizing AFs using within trust regions. The results in Figure 14 show that PR is a consistent best optimizer using TRs, but that STEs work quite well with TRs in many scenarios.

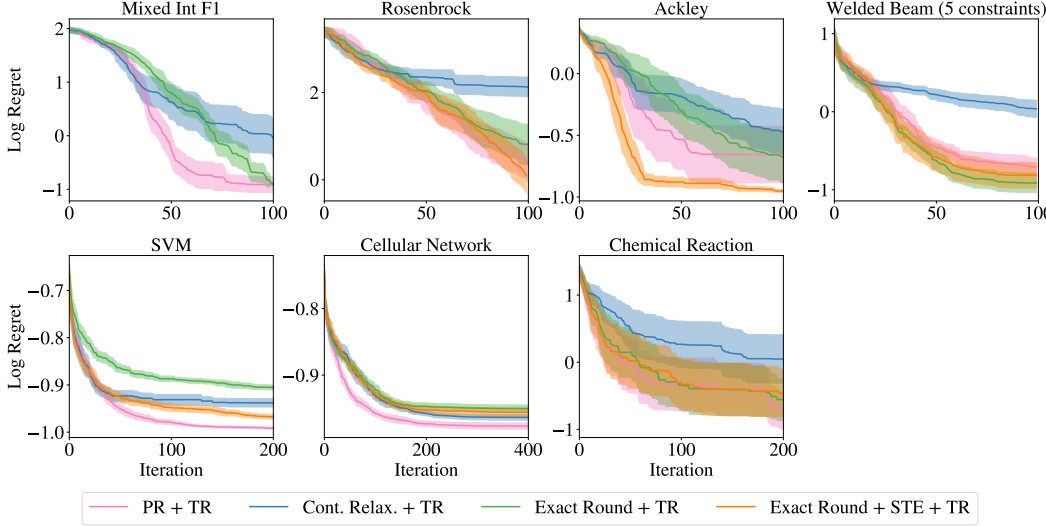

Figure 14: A comparison of TR methods with different acquisition optimization strategies. Log regret on each problem. We report log hypervolume regret for Oil Sorbent and report the log regret of the best feasible objective for Welded beam.

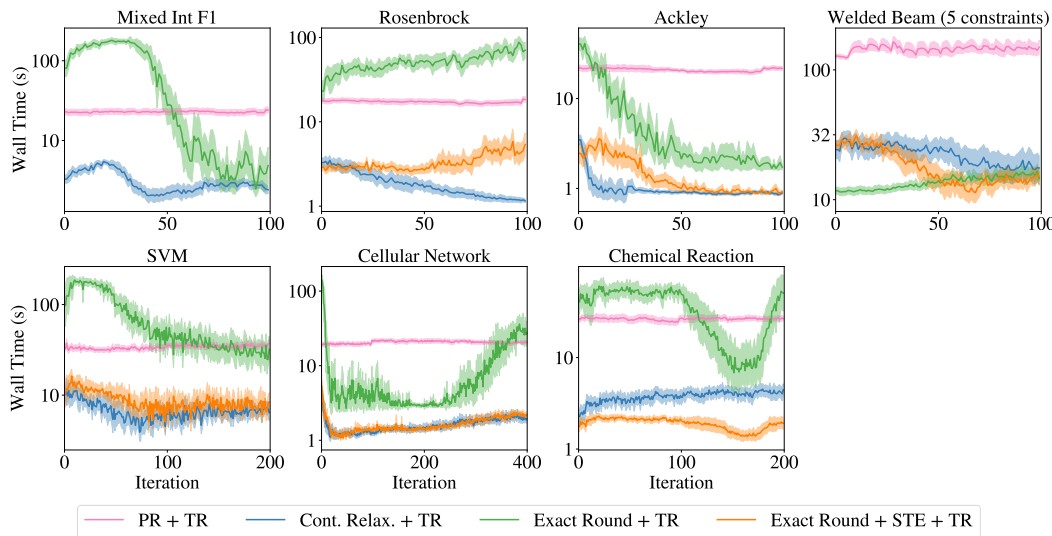

Figure 15: A comparison of wall times of TR methods with different acquisition optimization strategies.

# I Acquisition Function Optimization at a Given Wall Time Budget

In Figure 16, we provide additional starting points (64 points, rather than 20) to other non-PR methods in order to provide them with additional wall-time budget. We find that using PR with 64 MC samples, PR provides rapid convergence compared to other baselines and therefore is a good optimization routine for any wall time budget.

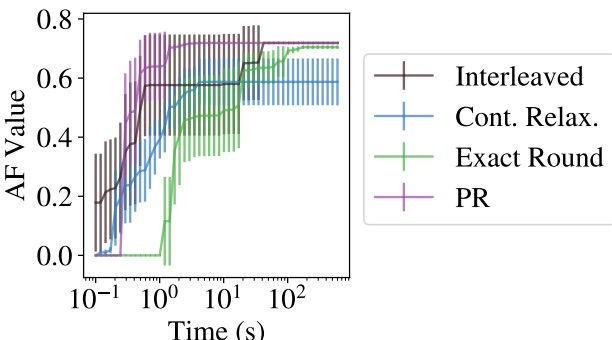

Figure 16: A comparison of methods for optimizing acquisition functions at a given wall time budget.

# J Alternative categorical kernels

In this section, we demonstrate that PR can be used with arbitrary kernels over the categorical parameters including those that require discrete inputs (which CONT. RELAX. is incompatible with). Specifically for the categorical parameters, we compare using (a) a Categorical kernel (default) versus with a Matérn-5/2 kernel with either (b) one-hot encoded categoricals, (c) a latent embedding kernel [67], or known embeddings based on density functional theory (DFT) [51]. For the latent embedding kernel, we follow Pelamatti et al. [44] and use a 1-d latent embedding for categorical parameters where the cardinality is less than or equal to 3 and a 2-d embedding for categorical parameters where the cardinality is greater than 3. For each latent embedding, we use an isotropic Matérn-5/2 kernel and use product kernel across the kernels for the categorical, binary, ordinal, and continuous parameters. For the kernel over DFT embeddings, we use the DFT embeddings for the direct arylation dataset from Shields et al. [51], which are available at https://github.com/b-shields/edbo.

It is worth noting that in the Chemical Reaction problem, the black-box objective is a GP surrogate model with a Categorical kernel that is fit to the direct arylation dataset. The purpose of this section is demonstrate that PR is agnostic to the choice of kernel over discrete parameters. Because the Chemical Reaction problem is based on a GP surrogate, we do not draw conclusions about which choice of kernel is best suited for modeling the true, unknown underlying Chemical Reaction yield function.

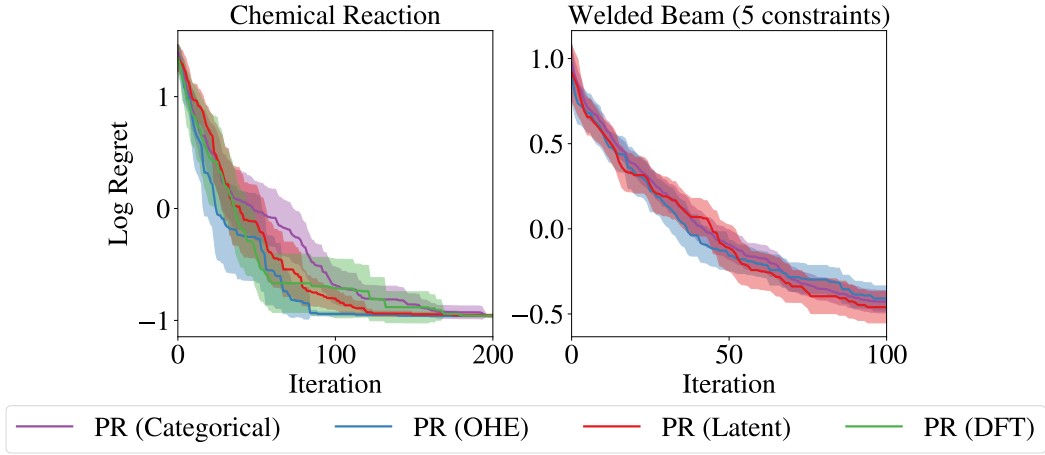

Figure 17: A comparison of different kernels over categorical parameters. Left: Welded beam has one categorical parameter, metal type (4 levels). Right: Chemical reaction has three categorical parameters, solvent, base, and ligand (with 4, 12, and 4 levels, respectively).

## K   Alternative Acquisition Functions

In this section, we compare PR with expected improvement (EI) against PR with upper confidence bound (UCB). For UCB, we set the hyperparameter $\beta$ in each iteration using the method in Kandasamy et al. [33]. Although UCB comes enjoys bounded regret [52], we find empirically that EI works better on most problems.

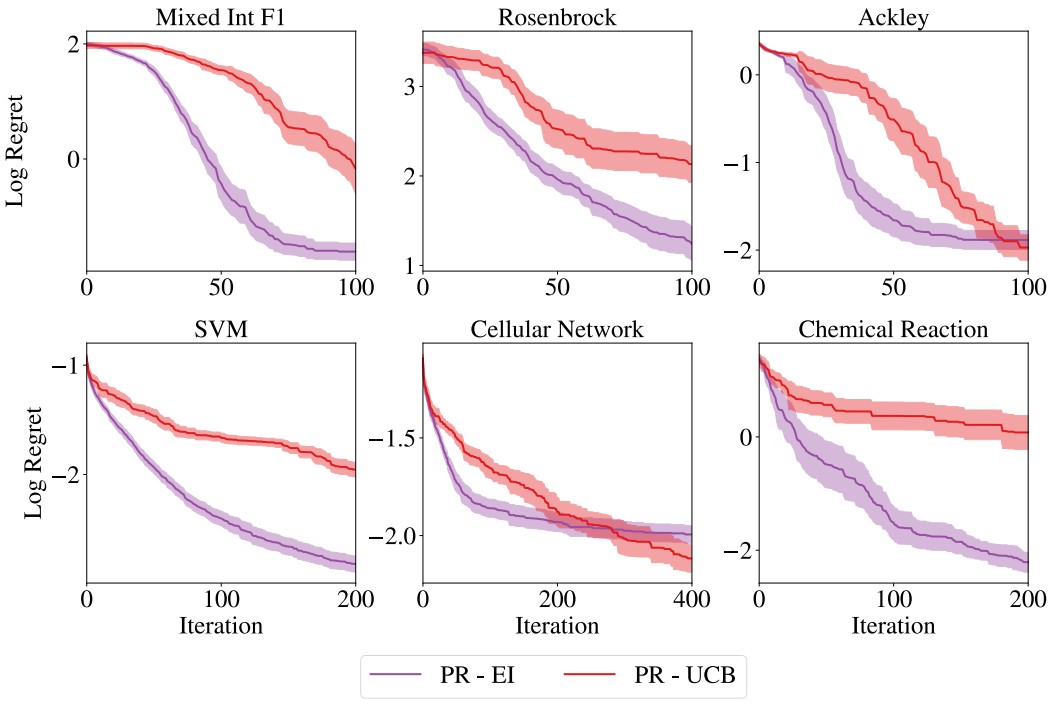

Figure 18: A comparison of expected improvement (EI) and upper confidence bound (UCB) acquisition functions with PR.

## L  Additional Results on Optimizing Acquisition Functions

In this section, we provide additional results on various approaches for optimizing acquisition functions using the same evaluation procedure as in the main text. We use 50 replications.

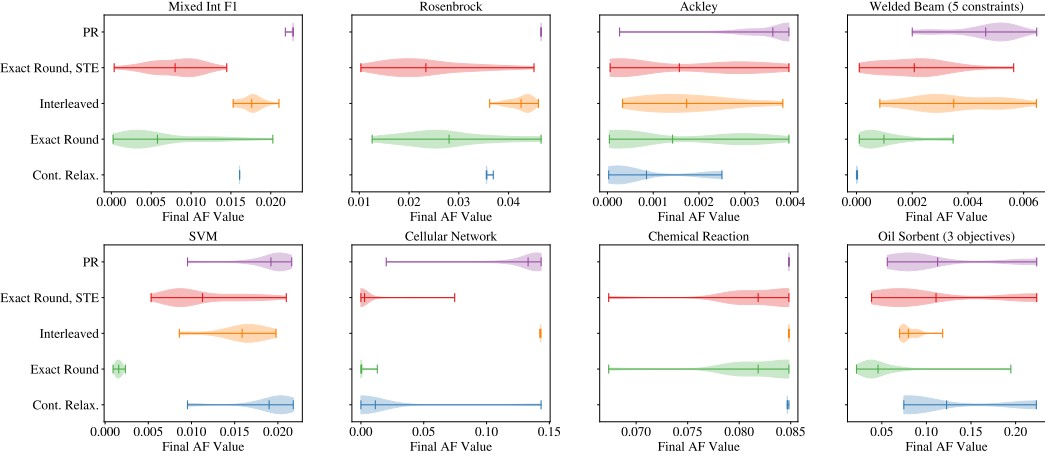

Figure 19: A comparison of methods for optimizing acquisition functions.

## M  Stochastic vs Deterministic Optimization

We compare optimizing PR with stochastic and deterministic optimization methods. For stochastic optimizers, we compare stochastic gradient ascent (SGA) and Adam with various initial learning rates. For SGA, the learning rate is decayed each time step $t$ by multiplying the initial learning rate by $t^{-0.7}$ and for Adam a fixed learning rate is used. For stochastic optimizers, the MC estimators of

PR and its gradient stochastic mini-batches of $N = 128$ MC samples are used. For deterministic optimization, base samples are kept fixed. All routines are run for a maximum of 200 iterations. In Figure 20, we observe that Adam is more robust to the choice of learning rate than SGA and generally is the best performing method. Furthermore, Adam consistently performs better than deterministic optimization. We compare Adam with a learning rate of $\frac{1}{40}$ against L-BFGS-B in Figure 21.

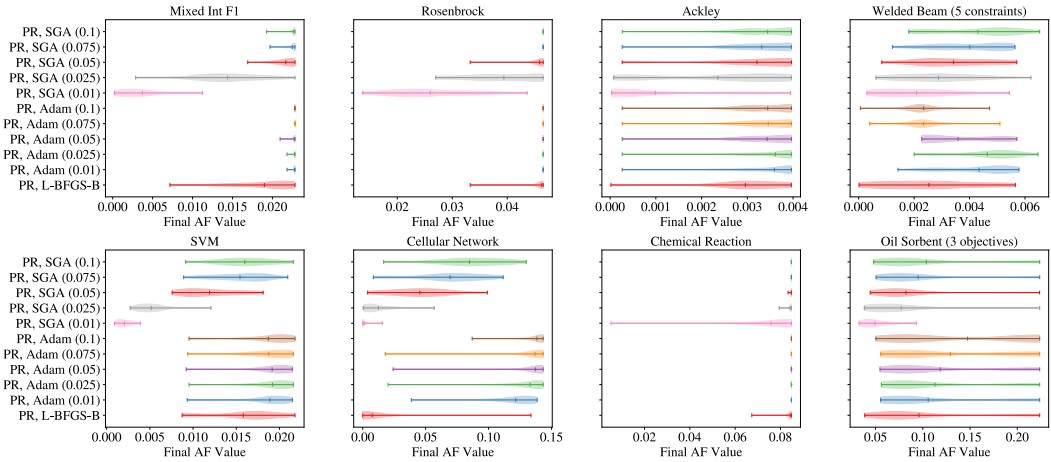

Figure 20: A comparison of PR using stochastic and deterministic optimization methods. The initial learning rate for stochastic gradient ascent is given in parentheses.

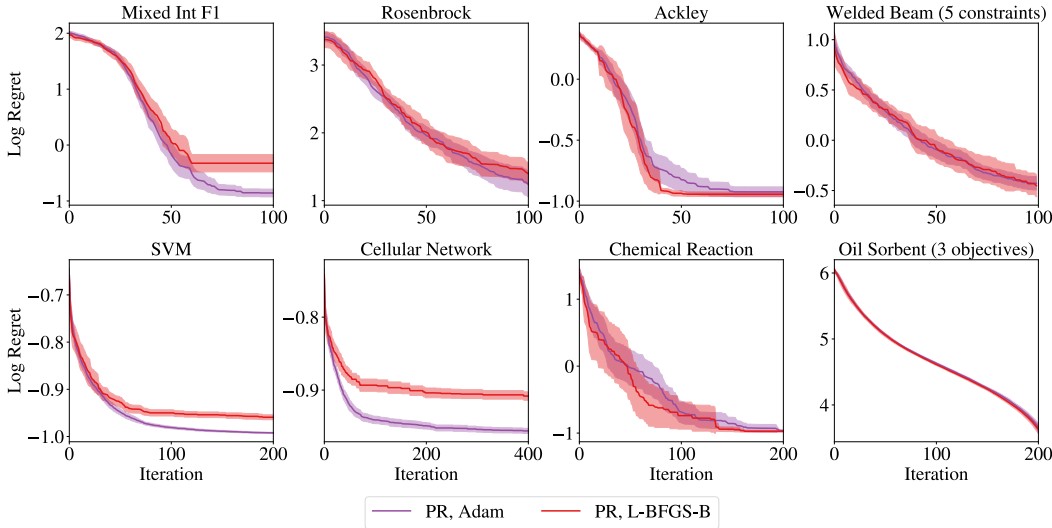

Figure 21: A comparison of optimizing the PO using deterministic estimation (via SAA) and optimization versus stochastic estimation and optimization.

# N    Comparison with an Evolutionary Algorithm

In Figures 22 and 22, we compare against the evolutionary algorithm PortfolioDiscreteOnePlusOne, which is the recommended algorithm for discrete and mixed search spaces in the Nevergrad package [45]. We find that PR significantly outperforms this baseline by a large margin with respect to log regret, but is slower than the evolutionary algorithm with respect to wall time.

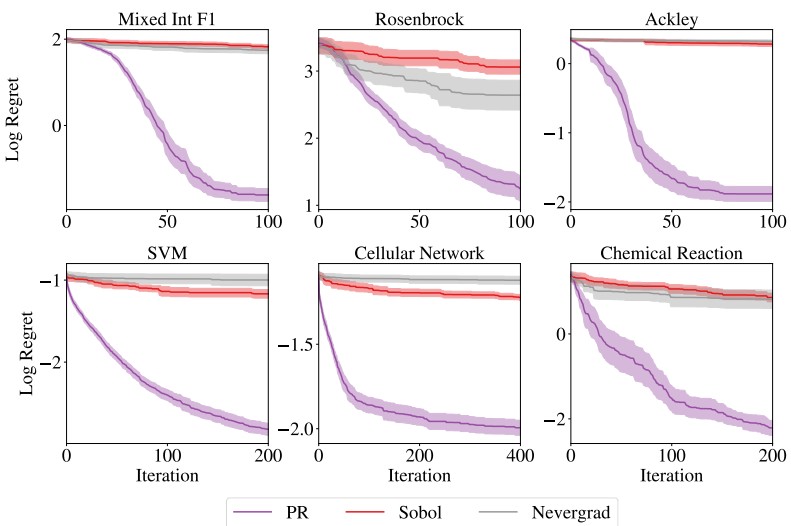

Figure 22: A comparison with an evolutionary algorithm with respect to log regret.

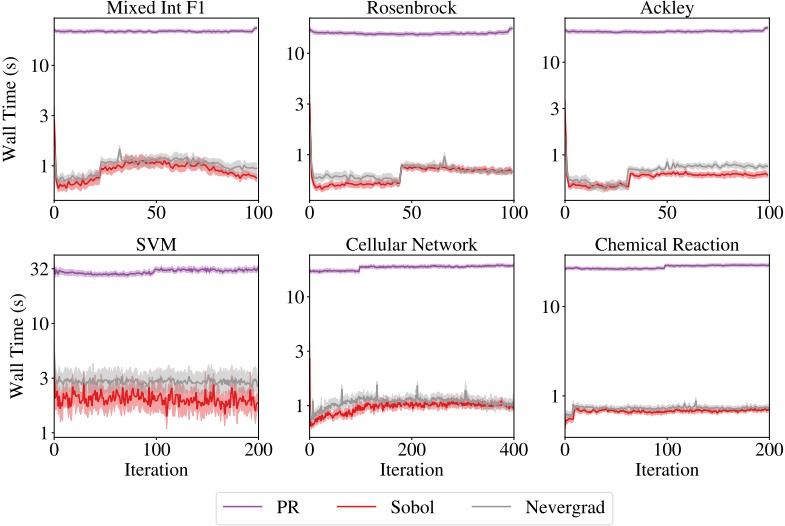

Figure 23: A comparison with an evolutionary algorithm with respect to wall time.