# OpenReview forum: "Bayesian Optimization over Discrete and Mixed Spaces via Probabilistic Reparameterization"
_NeurIPS.cc/2022/Conference — NeurIPS 2022 Accept_

### Official Review · Reviewer_H7wp · 2022-07-08

**Rating:** 9
**Confidence:** 4
**Soundness:** 4 excellent
**Presentation:** 4 excellent
**Contribution:** 4 excellent

**Summary:**

Maximizing Bayesian optimization acquisition function (AF) over mixed or high-cardinality discrete search spaces is challenging when the space is too large to enymerate directly. This is important because the theoretical regret bounds that come with certain AFs only apply if the AF can be maximized. The paper proposes to reparameterize the discrete parameters into probability distribution defined by continuous parameters, and then to maximize a probabilistic objective (PO) consisting of the expectation of the AF over those continuous parameters. The authors call this a probabilistic reparameterization (PR) of the AF. The paper then proves that maximizing this objective also maximizes the original AF, providing the same guarantees as maximizing the AF. They then derive unbiased and efficient Monte Carlo estimates of the PO and its gradients and show empirically that their method outperforms other mixed-variable optimizers on both maximizing the AF and the overall Bayesian optimization task.


**Questions:**

- In the middle and right panels of Figure 1, what does the color represent? I think it's the AF value, but in that case, why is it so different between the two panels?


**Limitations:**

The primary limitation is the computational cost of the MC estimation and maximization, which the authors address. This could be improved by explicitly stating the compute required for their experiments.

**Strengths And Weaknesses:**

## Strengths

- The idea of inducing a distribution with continuous parameters and then optimizing the expectation of the AF over the new parameters is both performant and elegant. It's the kind of solution that seems obvious once written down.
- The paper addresses an important limitation when applying Bayesian optimization to many (most?) real-world problems.
- The paper is well-contextualized in relation to previous work.
- The writing is generally clear, and the importance of the theorems is easy to see.
- The experiments seem sound, and the conclusions are supported by the results.

## Weaknesses
- I understand that there is limited space, but section 4.3 is very difficult to understand without a lot of prior knowledge.
- The experiments would be stronger with comparisons to non-BO black-box optimization methods, such as evolutionary algorithms or to methods that use eg a VAE to embed the discrete variables into a continuous space.

---

> ### Author Response · Authors · 2022-08-02
> **Response to H7wp**
>
> Thank you for your thorough review!  We address your specific comments below.
>
> > In the middle and right panels of Figure 1, what does the color represent? I think it's the AF value, but in that case, why is it so different between the two panels?
>
> ​​Correct. The reason is that a continuous relaxation over-estimates the AF value. The AF value at infeasible continuous points (red) is much higher than the best AF value at any feasible discrete point (white).
>
> > The experiments would be stronger with comparisons to non-BO black-box optimization methods, such as evolutionary algorithms or to methods that use, e.g., a VAE to embed the discrete variables into a continuous space.
>
> We have added Nevergrad’s recommended (Rapin et al., 2018) evolutionary strategy for mixed/discrete spaces (PortfolioDiscreteOnePlusOne) to our unconstrained single objective problems and provided the results in Appendix M in Figures 21 and 22. We observe that PR vastly outperforms Nevergrad on all test problems. We are happy to include alternative ES baselines if you have suggestions for better baselines.
>
> Regarding embeddings into a continuous space via VAEs: our method is compatible with any type of kernel, including embedding-based approaches.  For example, in Appendix H, we consider the embedding method of Zhang et al., Technometrics 2019.  There are a large variety of VAE methods that could be used for embedding the entire search space, each of which generally involves a problem-dependent kernel choice (see e.g., Kusner et al., ICML 2017, Dai et al., ICLR 2018, Stanton et al., ICML 2022).  With PR, one could use distances in the latent space, but optimize in the original discrete space.  This would avoid the VAE-BO issue of obtaining continuous latent designs from continuous optimization in the latent space that do not map to any discrete designs (the focus of many works—e.g. Griffiths et al., Chem Sci 2020 and Notin et al., NeurIPS 2021). Further work in this area is beyond the scope of this work.
>
> Kusner et al., ICML 2017. Grammar variational autoencoder.
>
> Dai et al., ICLR 2018. Syntax-directed variational autoencoder for structured data.
>
> Griffiths et al., Chem Sci 2020. Constrained Bayesian optimization for automatic chemical design using variational autoencoders.
>
> Notin et al., NeurIPS 2021. Improving black-box optimization in VAE latent space using decoder uncertainty.
>
> Rapin et al., Github 2018. Nevergrad - A gradient-free optimization platform.
>
> Stanton et al, ICML 2022: Accelerating Bayesian Optimization for Biological Sequence Design with Denoising Autoencoders
>
> > I understand that there is limited space, but section 4.3 is very difficult to understand without a lot of prior knowledge.
>
> Thank you for pointing this out. We have added a more detailed description in Appendix C.4 in our revision. We can move this to the main text in the camera ready when we are allowed an extra page, if you suggest.

---

> > ### Comment · Reviewer_H7wp · 2022-08-03
> > **Response to response**
> >
> > Thank you for addressing my minor concerns. My score remains the same.

---

### Official Review · Reviewer_EFXz · 2022-07-11

**Rating:** 5
**Confidence:** 5
**Soundness:** 2 fair
**Presentation:** 3 good
**Contribution:** 1 poor

**Summary:**

The paper addresses BO in mixed decision spaces, where part of the space is continuous and part is discrete. Namely, it focuses on the problem where the discrete space is large and cannot be exhaustively evaluated. The proposed strategy reparametrize the objective with a parametrized discrete probability distribution over the discrete decision space, which can be optimized using continuous optimization tools. Further extensions with MC sampling and extensive benchmark is provided.

**Questions:**

- In Fig. 1, AF evaluates the expectation exactly? I would not expect there to be no spread if used with monte-carlo sampling.
- Why not use Comsopolitan and HyBO on more realistic benchmarks? Could you not just add the constraint in the AF selection?
- Why on Ackley PR-TR performs better?
- Can in any of these experiments exhaustive search be performed? It would be nice to understand the effect of MC sampler.


**Limitations:**

1. The whole approach is sensible only if one of these two conditions are met:
a) mixed space. Let me denote the effort to optimize \mathcal{X} given a specific z to B. If B*|\mZ| is less than optimizing \mathcal{Z}, \mathcal{X} jointly with the Monte-Carlo or exact calculation
b) discrete space, but very efficient evaluation of MC sampling exists

My conjecture is that b) does not occur in practice much, and a) can be indeed practical.

2. Effect of MC sampling is not investigated in the experiments and in the motivating Figure.



**Strengths And Weaknesses:**

Strengths
----------
- I think this is a very practical algorithm if certain conditions are met (see Limitations). Authors identify problems where these conditions are met.
- solid empirical analysis beyond the standards of the BO field.
- optimization of AF is indeed an elephant in the room, nice to see that it is being recognized as a problem.

Weakness
---------
- I feel the contribution to the field does not warrant publication at the most prestigious venue. I think people were aware of the reparameterization trick in this context. The problem with it is that for discrete spaces its complexity is larger than evaluating the whole \mZ once. In mixed spaces, there are conditions where it could be better, but how much better is not addressed in this paper.
- The paper cites a lot of by now folclore theorems, such as unbiasedness of monte carlo, SGD convergence and regret rates of algorithms depending on AF. I think the paper could be explained in 1 page and experiments take 2 pages, exactly suitable for a workshop paper.
- The best performing algorithm PR+TR is not mentioned in the main text.
- Convergence of SGD on the non-convex objective is also not ideal, but I guess this is not addressed in the paper since the “discrete part” is the challenge. But its good to point out that by creating a continuous relaxation not all issues are removed.

---

> ### Author Response · Authors · 2022-08-02
> **Response to EFXz (1/3)**
>
> Thank you EFXz for your comments. We are surprised that you believe our contribution is poor (score of 1), despite acknowledging that “Optimization of AF is indeed an elephant in the room, nice to see that it is being recognized as a problem”. We are glad that you recognize that our method is practical for the common scenarios that we target in this work.
>
> > The problem with it is that for discrete spaces its complexity is larger than evaluating the whole [discrete search space] $\mathcal Z$ once.
>
> This depends entirely on the number of discrete configurations and number AF evaluations that PR uses during optimization. We find that PR is much faster than enumerating all discrete options on Welded beam (>370M discrete configurations), which has the only fully discrete search space of the problems in the paper. PR optimization (with 1,024 MC samples) of the PO took 250.9 seconds on average across 10 replications, whereas optimizing the AF by enumerating all discrete options took 1349.3 seconds.
>
> Nevertheless, in the setting where the AF can be evaluated at every discrete option in a reasonable amount of time, we would recommend to use enumeration rather than PR, since the black-box function is typically expensive to evaluate and time spent optimizing the AF is often comparably cheap. It is worth noting that enumeration is infeasible in many practical scenarios (more below) and on most problems in the paper. We can make this clearer in the discussion section.
>
> > In mixed spaces, there are conditions where [PR] could be better, but how much better is not addressed in this paper.
>
> Thank you for raising this. We have added results that include a new baseline method that enumerates all discrete configurations and optimizes the continuous parameters for each fixed discrete configuration to directly answer your question of “how much better”. It is only feasible to run this on chemistry with 192 discrete configurations and Ackley with 1024 discrete configurations. On both problems, we find that PR achieves the same optimizer as the enumeration baseline in a much shorter period of time (as shown in Appendix N in Figures 23 and 24 in our revision).
>
> > The whole approach is sensible only if one of these two conditions are met: a) mixed space. Let me denote the effort to optimize $\mathcal{X}$ given a specific $z$ to $B$. If $B\cdot|\mathcal Z|$ is less than optimizing $\mathcal{Z}$, $\mathcal{X}$ jointly with the Monte-Carlo or exact calculation b) discrete space, but very efficient evaluation of MC sampling exists
>
> Yes, thank you for noting this. The precise scenarios that our method is designed for are (a) mixed search spaces, (b) large discrete search spaces where enumeration would be extremely slow or infeasible. Note that the latter can occur rapidly as the number of discrete factors increases.
>
> > My conjecture is that b) does not occur in practice much
>
> In the real world, it is not uncommon to have expensive-to-evaluate combinatorial optimization problems with more factors than can be exhaustively evaluated. For example, consider the testing of compiler or database configuration changes. There are hundreds of parameters that can be tuned, and to test such changes out in production, time-intensive experiments must be conducted to understand the effects of these changes on metrics like CPU usage, memory usage, or latency [1]. In machine learning, models may contain dozens to hundreds of features, which might affect the runtime of algorithms (due to data retrieval or computation), in addition to the model loss [2]. The SVM feature selection problem is one toy example of this and includes 50 features ($2^{50}$ discrete configurations).
>
> [1] Design and Analysis of Benchmarking Experiments for Distributed Internet Services. Bakshy & Frachentberg. ACM Conference on the World Wide Web 2015.
> [2] Hidden Technical Debt in Machine Learning Systems. Sculley et. al. NeurIPS 2015.

---

> > ### Author Response · Authors · 2022-08-02
> > **Response to EFXz (2/3)**
> >
> > > Effect of MC sampling is not investigated in the experiments and in the motivating Figure.
> >
> > *This is incorrect*. In the motivating figure, MC sampling is used. As we mention in the second sentence of the Experiments Section (Sec. 6), “We use $N=1024$ MC samples in our experiments and demonstrate that PR is robust with respect to the number of MC samples (and compare against analytic PR, where computationally feasible) in Appendix F.”
> >
> > In particular, Appendix F.1 (Figures 5 and 6), includes a comparison of PR with varying numbers of MC samples against analytic PR on the chemical reaction and ackley problems, where computing analytic PR was feasible. We find that MC PR performs comparably with its analytic counterpart. We do not find statistically significant differences in optimization performance between 128, 256, 512, and 1024 samples vs analytic, but using 128 samples is approximately 8x faster than the 1024 samples used in our paper. For your convenience, we have provided the main experiments using PR and PR+TR with 128 MC samples in Figures 9 and 10 in Appendix F.1 in our revision. In our revision, we have added comparisons with 8,16,32, and 64 MC samples in Figures 7 and 8 in appendix F.1. We do see performance degradation with 64 or fewer samples, but the wall time is also much smaller.
> >
> > > Can in any of these experiments exhaustive search be performed? It would be nice to understand the effect of MC sampler.
> >
> > See our response above regarding comparisons with MC and analytic PR.
> >
> > > The paper cites a lot of by now folclore theorems, such as unbiasedness of monte carlo, SGD convergence and regret rates of algorithms depending on AF. I think the paper could be explained in 1 page and experiments take 2 pages, exactly suitable for a workshop paper.
> >
> > We respectfully disagree with you here. We include these results to be rigorous in our both empirical and theoretical evaluation. Theoretical results around the convergence of our estimators, as well as regret bounds are a component that many members of the community find valuable: similar theoretical results are key contributions of many other BO papers published at NeurIPS and ICML. Our main theoretical result (Theorem 1) is critical for showing that optimizing the probabilistic objective results in the same designs as the original mixed/discrete acquisition function optimization problem. This is valuable because therefore using PR yields the same BO policy as using the raw acquisition function, if optimized well.
> >
> > > In Fig. 1, acquisition function evaluates the expectation exactly? I would not expect there to be no spread if used with monte-carlo sampling.
> >
> > In Figure 1, Monte Carlo sampling is used. What spread are you referring to? If you are referring to the spread in final AF value, optimizing the probabilistic objective is using different fixed random base samples in each replication, so the optimization is not the same in each replication.
> >
> > > Why not use Camsopolitan and HyBO on more realistic benchmarks? Could you not just add the constraint in the AF selection?
> >
> > This is described in Sec 6.3. Both methods run on all problems (including real world problems like SVM, chemical reaction, cellular network) except welded beam (neither supports constraints in the current implementation) and oil sorbent (neither support multiple objectives). On Cellular Network, our submission included partial results from Casmopolitan on the cellular network problem, but Casmopolitan failed with numerical issues; we believe this may have to do with the trust region getting too small and Casmopolitan evaluating the same design repeatedly leading to ill-conditioned covariance matrices. Hence, complete results are not available for Casmopolitan. Similarly, HyBO also had ill-conditioned covariance matrix issues on the cellular network problem, so only partial results are included in our submission. On SVM (53d), HyBO is very slow because it scales poorly with the input dimension, and so we terminated HyBO reported partial results.
> >
> > > The best performing algorithm PR+TR is not mentioned in the main text.
> >
> > We do introduce PR+TR in the main text in Section 6. We will move a clearer description of the method (provided in the submission in Appendix C.2) to the main text in the camera ready version, when we are allowed an additional page.

---

> > > ### Author Response · Authors · 2022-08-02
> > > **Response to EFXz (3/3)**
> > >
> > > > Why does Ackley PR-TR perform better?
> > >
> > > PR-TR actually performs worse than PR on Ackley. Can you clarify your question please?
> > >
> > > In some problems such as cellular network, trust regions methods (e.g. Casmopolitan and PR-TR) are very good at zooming in on promising regions in the search space and perform better than non-trust region methods. In other cases where a GP is suitable surrogate model for the global landscape, non-trust region methods (e.g. standard BO) can outperform trust region methods like PR-TR such as on this ackley problem (this is also observed in Daulton et al., UAI 2022 in Figure 9).
> > >
> > > Daulton et al., UAI 2022. Multi-Objective Bayesian Optimization over High-Dimensional Search Spaces

---

> > ### Comment · Reviewer_EFXz · 2022-08-03
> > **discussion**
> >
> > I do not doubt that problems with a lot of discrete parameter exist and are relevant. But I doubt that doing MC sampling is addressed sufficiently in this work. Is it enough to do 1024 MC samples represent the expectation? How did you pick this number? Do you have some evidence/explanation for this? This is the crucial contribution of a work with this approach. Showing this is enough and why. If you were to show this then this is a good work. The idea in this paper is very simple and not groundbreaking but justification why this should work is of substance.
> >
> > I have a feeling one could come up with a counterexample, where reaching global optimum with such method with be very improbable.

---

> > > ### Author Response · Authors · 2022-08-03
> > > **Response to discussion (EFXz):**
> > >
> > > > But I doubt that doing MC sampling is addressed sufficiently in this work. Is it enough to do 1024 MC samples to represent the expectation?
> > >
> > > You may have missed our initial response regarding this question in the [2nd part (2 / 3)](https://openreview.net/forum?id=WV1ZXTH0OIn&noteId=RPsAHsgpRLz) of our response.  We performed a sensitivity analysis with respect the number of MC samples where we find that 128 works about as well as 1024 MC samples and **both perform comparably with computing PR analytically on the two small problems where that is feasible**. *The results are shown in Appendix F.1 (Figures 5, 6, 7, and 8).*
> > >
> > > To directly answer the question “Is it enough to do 1024 MC samples to represent the expectation?” We examine the MC approximation error relative to analytic PR on the chemistry (192 discrete configurations) and ackley problems (1024 discrete configurations). We have added the results in Figure 26 in Appendix P. These results report the mean absolute percentage error (MAPE)
> > >
> > > $\frac{100 }{|X_\text{discrete}|} \cdot \sum_{\boldsymbol x \in X_\text{discrete},\boldsymbol \theta \in \Theta_\text{discrete}} \frac{ \mathbb E_{\boldsymbol Z \sim p(\boldsymbol Z |\boldsymbol \theta)} \alpha(\boldsymbol x,\boldsymbol Z) - \frac{1}{N}\sum_{i=1}^N \alpha(\boldsymbol x, \boldsymbol z_i)}{\max_{\boldsymbol x \in X_\text{discrete},\boldsymbol \theta \in \Theta_\text{discrete}} \mathbb E_{\boldsymbol Z\sim p(\boldsymbol Z |\boldsymbol \theta)} \alpha(\boldsymbol x,\boldsymbol Z)}$
> > >
> > > evaluated over a random set of 10,000 points from $\mathcal X \times \mathcal \Theta$ (the sampled sets are denoted $X_\text{discrete}, \Theta_\text{discrete}$), where $\boldsymbol z_i$ are samples from $p(\boldsymbol Z |\boldsymbol \theta)$. We observe a rapid reduction in MAPE as we increase the number of samples. With 1024 samples, MAPE is 0.055\% (+/- 0.0002 \%) over 20 replications (different MC base samples in PR) on the chemical reaction problem and MAPE is 0.018\% (+/- 0.0003 \%) on the Ackley problem.   With 128 samples, MAPE is 0.282\% (+/- 0.0029 \%) over 20 replications (different MC base samples in PR) on the chemical reaction problem and MAPE is 0.052\% (+/- 0.0021 \%) on the ackley problem. We find in our closed loop optimization experiments that these error rates are small enough to achieve SoTA BayesOpt performance, while still remaining computational feasible.
> > >
> > > > How did you pick this number? Do you have some evidence/explanation for this?
> > >
> > > We initially just chose 1024 because our intuition was that a large number of MC samples would be needed, but in a sensitivity analysis we performed a few days prior to our submission (the one reported in Appendix F.1), we found that smaller numbers of MC samples, including 128 samples, does not degrade optimization performance, even on the problems with search spaces with high dimensionality and many discrete configuration (such as SVM).  See detailed results in Appendix F.1.
> > >
> > > > This is the crucial contribution of a work with this approach. Showing this is enough and why. If you were to show this then this is a good work.
> > >
> > > We believe we have provided strong evidence in response (2/3) and above regarding the approximation error of MC sampling. Please let us know if there are any additional experiments or explanations to convince you further.
> > >
> > > Another interesting data point is regarding the rank order of MC vs analytic PR under sample average approximation. For example, if analytic PR has higher value for some $\boldsymbol x_1, \boldsymbol \theta_1$ than a different $\boldsymbol x_2, \boldsymbol \theta_2$, what is the probability that the MC estimator of PR gives higher value to the better design according to analytic PR? Using 10,000 random pairs of points from $\mathcal X \times \Theta$, we find that the MC estimator of PR gives higher value to the $\boldsymbol x_1$, $\boldsymbol \theta_1$ with probability 0.92.

---

> > > > ### Comment · Reviewer_EFXz · 2022-08-03
> > > > **discussion**
> > > >
> > > > How is your MC scheme different to just picking 1024 (or MC numbers) random configuration and calculating the best out of those? Aren't you effectively randomly subsampling configurations and picking the best among those? Why this probabilistic interpretation is needed?

---

> > > > > ### Author Response · Authors · 2022-08-03
> > > > > **response 2 to EFXz on MC sampling**
> > > > >
> > > > > > How is your MC scheme different to just picking 1024 (or MC numbers) random configuration and calculating the best out of those?
> > > > >
> > > > > In some BO papers, acquisition functions are optimized over a discrete set of inputs. E.g. sample some finite set of values $X$ (a sample from the whole search space) and the next point is selected by maximizing the acquisition function over this discrete set. This is not what we are doing. Our approach is completely and fundamentally different. Please let us know if this is confusing and we will elaborate further.
> > > > >
> > > > > > Aren't you effectively randomly subsampling configurations and picking the best among those? Why is this probabilistic interpretation needed?
> > > > >
> > > > > Rather than sampling some discrete configurations $\boldsymbol z$ from the search space and taking the best, we have reparameterized the problem by using a random variable $\boldsymbol Z$ governed by $\boldsymbol \theta$. With 1024 MC samples, we sample 1024 configurations of $\boldsymbol Z$ from the distribution defined by $\boldsymbol \theta$ at each step of optimizing the probabilistic objective—many such steps occur to find the optimal $\boldsymbol \theta$. *We optimize $\boldsymbol \theta$ with gradients, so $\boldsymbol \theta$ changes during the continuous optimization and thus the samples of $Z$ also change.* Note that this is fundamentally different from sampling configurations $\boldsymbol z$ from the search space and taking the best.
> > > > >
> > > > > Does this answer your question? Did our other responses in this thread and in https://openreview.net/forum?id=WV1ZXTH0OIn&noteId=RPsAHsgpRLz you address your other concerns?
> > > > >
> > > > > Thanks!

---

> > ### Comment · Reviewer_EFXz · 2022-08-05
> > **response**
> >
> > So let me take the example of M binary random variables from line 110. Then using (1) we construct the reparametrization. There are exactly as many theta variables (M) as there are binary random variables. In order to evaluate (1), we need to sum over 2^M variables.
> >
> > In reasonable application such as M=50 as you mention above 2^M is really not feasible. So you resort to MC sampling and sample N configurations $\tilde{z}_i$ to construct a gradient w.r.t theta using the "log trick" or "policy gradients." The same gradient would arise if I define my function just on point estimates.
> >
> > Using Theorem 1, which I find obvious, one can show actually that, if the samples $\tilde{z}_i$ are always the same in each round of optimization you are identifying just the best configuration out of $\tilde{z}_i$ effectively the same as if you subsampled your discrete space. This is exactly the point identified by the reviewer mBPd. In appendix C.4, you actually say that you always use the same $\tilde{z}_i$ or another way to keep the same stochasticity fixed. I think you approach could work and would be different to subsampling discrete sample if at every step you chose a different random $\tilde{z}_i$. Then this would reduce to a non-convex stochastic optimization problem like encountered with SGD on neural nets.

---

> > > ### Author Response · Authors · 2022-08-05
> > > **Response 3**
> > >
> > > Dear reviewer,
> > >
> > > We appreciate your engagement, but there is clearly a misunderstanding here. Thanks in advance for working together to try to resolve this gap with us.
> > >
> > > > In order to evaluate (1), we need to sum over 2^M variables.
> > >
> > > We need to sum over 2^M terms, not variables. Theorem 1 proves that the maximizers of PR are consistent with those of the original AF when all combinations can be enumerated and we show in Theorem 2 that our MC estimator is unbiased.  Our method is devised precisely for the case where not all terms can be enumerated, and we empirically show that this MC estimator converges rapidly and produces SoTA results in practice.
> > >
> > > > In reasonable application such as M=50 as you mention above 2^M is really not feasible. So you resort to MC sampling and sample N configurations  to construct a gradient w.r.t theta using the "log trick" or "policy gradients."
> > >
> > > A main contribution of the paper is to show that this gradient of the probabilistic objective w.r.t. theta, while analytically infeasible to compute, can be estimated via MC the likelihood ratio method (a.k.a REINFORCE).  We have provided direct empirical evidence for this in the original submission (Appendix F.1), in our [responses](https://openreview.net/forum?id=WV1ZXTH0OIn&noteId=7rz0tyJR2mZ) / updated manuscript
> > >
> > > > The same gradient would arise if I define my function just on point estimates.
> > >
> > > We don’t understand this. Can you please clarify?
> > >
> > > > ​​Using Theorem 1, which I find obvious, one can show actually that, if the samples $\tilde{z}_i$ are always the same in each round of optimization you are identifying just the best configuration out of effectively the same as if you subsampled your discrete space.
> > >
> > > This is incorrect. The samples $\tilde{z}_i$ are not the same in each round of optimization.
> > >
> > > Based on your comment “I think you approach could work and would be different to subsampling discrete sample if at every step you chose a different random $\tilde{z}_i$. Then this would reduce to a non-convex stochastic optimization problem like encountered with SGD on neural nets”, you seem to understand the idea of our method under resampling the samples $\tilde{z}_i$ at each step and using stochastic optimization. **Now, crucially, the $\tilde{z}_i$ are different in each iteration**. However, we don’t use random resampling + SGD (this would also be an option), but instead use an sample average approximation approach (otherwise known as the “method of common random numbers”) to render the optimization deterministic (conditional on a single draw of random base samples).
> > >
> > > You are correct that we fix **some** random variables to reduce the variance of our MC estimators. However, **WE DO NOT FIX the samples $\tilde{z}_i$.** These samples vary as $\theta$ varies during the optimization! This is of critical importance. Instead of fixing the samples $\tilde{z}_i$ during the optimization, we fix *uniform base samples* using the reparameterization in Table 4. This is described in Appendix C.4. Although these base samples are fixed, the samples $\tilde{z}$ will indeed vary as $\theta$ varies. Is this clear? This point is very important. A similar techinique is used for variance reduction in Sec 4 of [1] and is the standard way of optimizing MC acquisition functions in [BoTorch] (https://github.com/pytorch/botorch) under a different reparameterization of base samples.
> > >
> > > > This is exactly the point identified by the reviewer mBPd. In appendix C.4, you actually say that you always use the same \tilde{z}_i or another way to keep the same stochasticity fixed.
> > >
> > > This is a misunderstanding. We fix the uniform base samples, not the samples $\tilde{z}_i$. This means that $\tilde{z}_i$ can (and does) vary while optimizing the probabilistic objective. Reviewer mBPd appears to also have an incorrect interpretation of our results, which we clarified in [our response](https://openreview.net/forum?id=WV1ZXTH0OIn&noteId=vyEpCV7amUF)
> > >
> > > We’d also like to note that SAA with L-BFGS-B outperforms the stochastic gradient ascent approach (Appendix K), and is free from hyperparameters such as learning rates or momentum.
> > >
> > > **Finally, in your initial review, you noted:**
> > >
> > > > But I doubt that doing MC sampling is addressed sufficiently in this work. Is it enough to do 1024 MC samples represent the expectation? How did you pick this number? Do you have some evidence/explanation for this? This is the crucial contribution of a work with this approach. Showing this is enough and why. If you were to show this then this is a good work.
> > >
> > > Did our prior responses and additional analysis address this question for you?
> > >
> > > [1] Balandat, M., et. al. (2020). BoTorch: a framework for efficient Monte-Carlo Bayesian optimization. Advances in neural information processing systems, 33, 21524-21538.

---

> > > > ### Comment · Reviewer_EFXz · 2022-08-08
> > > > **response**
> > > >
> > > > - Thank you. I understand better the flaw I thought is not present. I will raise my score.
> > > >
> > > > - I also checked your experiments in F.1 and there is some evaluation to the number of MC samples. So indeed you performed this check, which is nice. This makes it well-rounded practical evaluation.
> > > >
> > > > - I have to say I am not fond of the Theorems 1 and 2, since they are folklore and you should either cite them as *direct* corollaries or not state at all. I do not think these are statements worth calling theorems. Also in Thm.2, Robbins-Monro conditions require not only the square of the stepsizes to be finite but also the sum of step-sizes to diverge - so you are missing one of the conditions to the best of my knowledge, but well this is easily fixable so I am not reducing points for this.
> > > >
> > > > My belief is still that this has decent practical benchmark but this reparametrization trick is not very deep result or insight. I would really love to see this subsampling benchmark where one subsamples the search space randomly iid, with the same number of MC \times iteration numbers and compare to that. I hardly doubt it would be competitive. I think there is no free lunch here. If you have a good reason, not anectodal like the experiments performed why this is always better, I am all ears.  Anyway, I think there is no harm in accepting this paper.

---

### Official Review · Reviewer_mBPd · 2022-07-11

**Rating:** 5
**Confidence:** 3
**Soundness:** 3 good
**Presentation:** 4 excellent
**Contribution:** 3 good

**Summary:**

This work proposes a method for optimization of acquisition functions in Bayesian optimization (BO), where at least some of the input variables are discrete (binary, integer, or categorical). The method is based on a probabilistic reformulation, whereby the discrete variables are cast as random variables, drawn from probability distributions. Optimization can then performed on the expected value of the acquisition function, using the parameters of the assumed probability distributions as inputs instead of the original (discrete) variables. This reformulation enables the use of gradient-based optimization; the authors propose a Monte Carlo approach to estimate the gradients.

**Questions:**

Could a comparison be added to show that the proposed MC method results in more sample-efficient (in terms of optimization) performance than existing methods? E.g., when 1024 MC samples are used, would it be a more fair comparison to allow the multi-start methods to have 1024 starts?

Since the proposed method effectively updates continuous parameterizations used to compute the "discretized" values of the original updates, is there an equivalent relaxation and iterative optimization method in the original space?

**Limitations:**

Yes, provided in supplementary material.

**Strengths And Weaknesses:**

STRENGTHS:

1. The method and significance are well presented, and the introduction and case studies show why the setting of BO over mixed variable spaces is both challenging and important.

2. The computational studies demonstrating the method are very thorough, with a wide range of case studies and tests considered.

3. The authors prove mathematically that the proposed reformulation does not affect the convergence guarantees of standard BO.

WEAKNESSES:

1. I am not very familiar with probabilistic reformulation, but it seems the proposed reformulation effectively inherits the weaknesses of BO on discrete inputs that it purports to avoid. For example, a binary variable is recast as a binomial distribution, which must be again "discretized" (in Algorithm 1) to create feasible values for $z_n$. Moreover, the secondary transformation in Table 3 introduces similar techniques used to relax discrete variables for optimization.

2. While the main advantage of the proposed method is that it enables gradient based optimization, the accuracy of the gradients relies on Monte Carlo estimation, which must be evaluated over the original discretized space (eg. 6). This introduces the same curse of dimensionality that would arise from optimization over a mixed variable space (e.g., by enumeration or branch-and-bound), and, as a result, the computational results may be unfair as the proposed method is given much more information in the form of MC samples.

---

> ### Author Response · Authors · 2022-08-02
> **Response to mBPd (1/2)**
>
> Thank you for your thoughtful review and questions! We are glad to hear that you found the soundness and contribution of our work to be good (scores of 3) and the presentation to be excellent (score of 4). We hope that this response will clarify crucial aspects of the paper to change your score from a borderline reject to an accept.
>
> > I am not very familiar with probabilistic reformulation, but it seems the proposed reformulation effectively inherits the weaknesses of BO on discrete inputs that it purports to avoid. For example, a binary variable is recast as a binomial distribution, which must be again "discretized" (in Algorithm 1) to create feasible values for $z_n$. Moreover, the secondary transformation in Table 3 introduces similar techniques used to relax discrete variables for optimization.
>
> The proposed approach does not inherit the weakness you describe, and this is the main theoretical result of the work. In particular, Theorem 1 of our algorithm is that if an optimal $\boldsymbol \theta_n$ is found, then any discrete sample $\boldsymbol z_n  ~ p(\boldsymbol Z|\boldsymbol \theta_n)$ is optimal and that for any optimal $\boldsymbol z_n$ there is a $\boldsymbol \theta_n$---that assigns nonzero probability to $\boldsymbol z_n$— that is optimal with respect to the probabilistic objective. Theorem 1 says that ($\boldsymbol x_n, \boldsymbol z_n$) is guaranteed to be optimal with respect to the acquisition function. In the case of a unique best optimizer (here we consider the case of a single discrete parameter for simplicity), $\theta_n$ is a point mass on the best $z_n$, in which case discretize in Algorithm 1 is simply to map a one-hot categorical to a categorical representation of $[0, C-1]$. If there are multiple $z_n$ that are optimal for a given optimal $x_n$, then if the optimal discrete values are consecutive ordinals or $z_n$ is binary or categorical, then $\theta_n$ may not be a point mass, but rather provide support exclusively over optimal values of $z_n$. If the $z_n$ is ordinal and the optimal values are not consecutive, then there could be multiple $\theta_n$ that point masses on the different optimal $z_n$.  In contrast, with a continuous relaxation, the optimal continuous relaxation $z'^*$ does not necessarily lead to an optimal $z_n$ when $z'^*$ is rounded. In our revision we have provided an updated theoretical formulation  that should help clarify some of these points.
>
> > Moreover, the secondary transformation in Table 3 introduces similar techniques used to relax discrete variables for optimization.
>
> Particularly in response to this statement, the reparameterization in Table 3 is used for computational stability and is commonly used (e.g., Yin et al., 2020). Table 3 reparameterizes the parameter $\theta$ of a discrete probability distribution,  and the probability distribution still only provides support on discrete values $z$. Therefore, this reparameterization of $\theta$ *does not* lead to the aforementioned overestimation issue with continuous relaxations of discrete parameters with AFs because the AF is only evaluating on discrete values sampled from $p(z|\theta)$.

---

> > ### Author Response · Authors · 2022-08-02
> > **Response to mBPd (2/2)**
> >
> > > While the main advantage of the proposed method is that it enables gradient based optimization, the accuracy of the gradients relies on Monte Carlo estimation, which must be evaluated over the original discretized space (eg. 6). This introduces the same curse of dimensionality that would arise from optimization over a mixed variable space (e.g., by enumeration or branch-and-bound), and, as a result, the computational results may be unfair as the proposed method is given much more information in the form of MC samples.
> >
> > The MC approximation error increases with the dimension of the search space, but the cost is fixed for a fixed number of MC samples regardless of the dimension of the search space. Enumeration would scale poorly (time wise) with the dimension of the search space, whereas PR is scalable and feasible (see results on enumeration in the general discussion to all reviewers).
> >
> > As to the computational results being “unfair” since the “proposed method is given much more information”, we are not sure we understand this point. All methods utilize the same number of costly evaluations of the underlying black box function. The different approaches indeed evaluate the acquisition function in different ways, which affects not only the optimization performance w.r.t the black-box optimization task, but also computational complexity and runtime of the AF. We address these tradeoffs explicitly in the paper. We hope that our response to your review and the others will also help clarify the substantive differences in runtime even more.
> >
> > > E.g., when 1024 MC samples are used, would it be a more fair comparison to allow the multi-start methods to have 1024 starts?
> >
> > If we are considering the cost of optimizing the AF, then giving more starting points to non-PR methods would help equalize the computational budget for all methods. However, PR is not very sensitive to the number of MC samples (see Figures 5-8 in Appendix F.1), and the number of MC samples can be reduced significantly (from 1024 to 128) to improve wall time without degrading optimization performance. PR with 128 MC samples still outperforms alternatives (see Figures 9-10 in Appendix F.1).
> >
> > To further provide a demonstration of AF optimization performance at a particular wall time budget, we provided additional starting points to non-PR methods (64 instead of the default 20) and compared against PR with 64 MC samples. The results in Figure 25 in Appendix O in our revision show that PR outperforms alternatives on the chemistry problem regardless of wall time budget.
> >
> > > Could a comparison be added to show that the proposed MC method results in more sample-efficient (in terms of optimization) performance than existing methods?
> >
> > Please see our experiments in Section 6 (summarized in Fig. 2), which demonstrates that our method is more sample-efficient than other methods, in terms of optimization of the expensive-to-evaluate black-box function.

---

> > ### Comment · Reviewer_mBPd · 2022-08-06
> > **Discussion**
> >
> > Thanks for the additional explanations about the reformulation. The provided examples clarify how the formulation results in a different problem compared to relaxation strategies. I will update my rating accordingly. Regarding MC sampling, the results presented indeed show that good results can be attained on the problems considered in this paper with relatively few samples, but this paper could be strengthened with additional theory that this holds true for the general case (or under a certain set of assumptions).

---

> ### Author Response · Authors · 2022-08-06
> **Check in with mBPd**
>
> Dear mBPd,
>
> We hope you are well. We have clarified all misunderstandings (including a crucial aspect of our work) and addressed your concerns. Do you have any remaining concerns after reading our response? We believe that your score does not reflect your review, especially given our rebuttal, and we kindly request that you consider increasing your score.
>
>  Thank you!

---

### Official Review · Reviewer_9pbz · 2022-07-12

**Rating:** 7
**Confidence:** 4
**Soundness:** 3 good
**Presentation:** 3 good
**Contribution:** 3 good

**Summary:**

This paper proposes a Bayesian optimization method over mixed spaces, i.e., discrete space and continuous space, via probabilistic reparameterization.  Since optimizing an acquisition is a challenging task where discrete variables exist, probabilistic reparameterization is used to allow us to optimize the acquisition function with gradient information.  The authors provide the theory that the proposed method serves a consistent optimizer.  Finally, they demonstrate that the proposed method works well in diverse experiments.

**Questions:**

I would like to ask the authors some questions.

1. In Table 1, ordinal variables have the form of continuous relaxation $z' \in [0, C - 1]$ and categorical variables have the form of continuous relaxation $z' \in [0, 1]^C$.  Are they correct?  For ordinal variables, $[0, 1) \to 0, [1, 2) \to 1, \ldots [C-2, C-1) \to C-2, [C-1, C-1] \to C-1$.  Thus, $C-1$ has a smaller range than the other integers.  I think it should be $[0, C)$.  Moreover, for categorical variables, $[0, 1]^C$ should be a simplex.  More precisely, a probability vector for categories has to sum to one, so I think it should be $[0, 1]^{C-1}$ where a sum-to-one constraint is assumed.  It is a generalization of binary case, which implies that its continuous relaxation is $[0, 1]^1$.

1. I am not sure if the continuous relaxations are changed, other parts are still same.  Or the current version is just equivalent to the continuous relaxations I described.  If updates are required, please describe in the rebuttal.

**Limitations:**

I do not think that this work has any negative societal impacts and any specific limitations.

**Strengths And Weaknesses:**

## Strengths

+ It solves an interesting topic regarding discrete and mixed spaces.

+ It proposes a solid method using probabilistic reparameterization.

+ The experimental results are reasonable.

## Weaknesses

- I think this work is quite novel, but some previous work such as [20, 50] should be discussed more thoroughly.

Please see the text box described below.

---

> ### Author Response · Authors · 2022-08-02
> **Response to 9pbz**
>
> Thank you for your review! We are glad that you found our work to be “quite novel.” The only weakness you listed was to discuss [20,50] more thoroughly. Given that this and your question regarding bounds for the continuous relaxation were your only concerns (both of which we address below), we hope that this will resolve any ambiguity and make you consider increasing your score.
>
> > 1. In Table 1, ordinal variables have the form of continuous relaxation $z' \in [0,C−1]$ and categorical variables have the form of continuous relaxation $z' \in [0,1]^C$. Are they correct?
> > 2. For ordinal variables, [0,1)→0,[1,2)→1,…[C−2,C−1)→C−2,[C−1,C−1]→C−1. Thus, C−1 has a smaller range than the other integers. I think it should be [0,C). Moreover, for categorical variables, $[0,1]^C$ should be a simplex. More precisely, a probability vector for categories has to sum to one, so I think it should be $[0,1]^C$ where a sum-to-one constraint is assumed. It is a generalization of the binary case, which implies that its continuous relaxation is [0,1].
> > 3. I am not sure if the continuous relaxations have changed, other parts are still the same. Or the current version is just equivalent to the continuous relaxations I described. If updates are required, please describe in the rebuttal.
>
> The discretize()  function (Table 1) actually makes it so that values in $[0, 0.5)$ are rounded to $0$ and values in $[C-1-0.5, C-1]$ are rounded to $C-1$. One could simply choose to use the bounds $[-0.5, C-0.5)$ as the bounds for the continuous relaxation. If you look at the code we provided, we already use $[-0.5, C-0.5)$ for the Sobol baseline and the Sobol initialization for all methods. We have updated this in our revision. For numerical optimization of the continuous relaxation, using either set of bounds should yield comparable performance. In fact, expanding the bounds seems like it would only hurt the performance of the continuous relaxation baseline since the additional space will remain unexplored.
>
> For a continuous relaxation, the vector of values across categories is not a probability distribution and does not need to sum to 1. This is the same relaxation used in [20]. For PR, the $\theta$ is a probability vector that does need to sum to 1. Thanks for pointing this out; we have clarified this in Table 2. We emphasize that using the transformations proposed in Table 3, $\theta$ is always in the simplex.
>
> > some previous work such as [20, 50] should be discussed more thoroughly
>
> [20] is the work by Garrido-Merchan et al. that proposes the ExactRound method, which we discuss at the end of Section 2 (L97-L101 in the original manuscript) and in Section 5 (L255-258 in the original manuscript). In our revision, we have made it clearer in Section 2 that [20] avoids the over-estimation issue with the continuous relaxation by discretizing the continuous relaxation before optimizing the acquisition function. However, applying discretization before evaluating the acquisition function makes the acquisition function non-differentiable with respect to the continuous relaxation of the discrete parameters. Therefore, Garrido-Merchan et al. rely on using finite differences to approximate the gradients of the acquisition function, which has large flat regions across slices of the continuous relaxation.
>
> [50] Is the work by Wilson et al. on using the reparameterization trick to approximate the expectation over the GP posterior in many acquisition functions via Monte Carlo. We mention this briefly in Section 5 (L266-268 in the original manuscript). While this reparameterization trick has some similarities with our probabilistic reparameterization, [50] reparameterizes an *existing* multivariate normal random variables in terms of standard normal random variables and then proposes to use a sample-path gradient estimator. In contrast, our approach *introduces* a new probabilistic formulation using discrete probability distributions and uses likelihood-ratio-based gradient estimators since sample-path gradients cannot be computed through discrete sampling. We have added this discussion to Appendix L in our revision and we will move this to the main text in the camera ready when we are permitted a 10th page.

---

> > ### Comment · Reviewer_9pbz · 2022-08-03
> > **Thank you for your comment**
> >
> > Thank you for your comment!
> >
> > After reading your comment and the revision, I was able to understand your work much clearer.
> >
> > I will increase the score slightly.

---

### Author Response · Authors · 2022-08-02
**Response to all reviewers**

Dear reviewers,

Thank you for your time and for the thorough feedback. We appreciate that you all recognized that our work is novel and solves an “interesting” (9pbz), “challenging and important” (mBPd) topic of optimizing acquisition functions over discrete and mixed spaces in Bayesian optimization, which is common in “many real-world problems” (H7wp). As EFXz articulated, “optimization of acquisition function is indeed an elephant in the room, nice to see that it is being recognized as a problem.”

We proposed using a theoretically-grounded probabilistic reparameterization (PR) of the discrete parameters in terms of discrete random variables, governed by continuous parameters. The gradients with respect to the continuous parameters can be computed analytically or estimated via unbiased Monte Carlo estimators to enable efficient optimization of the probabilistic objective, the expectation over the introduced probability distributions. H7wp found this approach to be “performant and elegant” and remarked that “It's the kind of solution that seems obvious once written down.”

Our work includes a “solid empirical analysis beyond the standards of the BO field” (EFXz). Furthermore, we proved “mathematically that the proposed reformulation does not affect the convergence guarantees of standard BO”, and we appreciate that to a reader “the importance of the theorems is easy to see” (H7wp).

Our work has broad impact for this “challenging and important” problem space because probabilistic reparameterization is agnostic to the choice of acquisition function and model (we demonstrate in Appendix H and I respectively), it is compatible with black-box constraints (see Welded Beam problem, Section 6) and multiple objectives (see Oil Sorbent problem, Section 6), and it is complementary with many methods including TuRBO (Section 6) (Eriksson et al., NeurIPS 2019).

Unanimously, reviewers found our writing to be clear (presentation scores of 3,3,4,4) and remarked that our method was sound (soundness scores of 3,3,3,4). And 3/4 reviewers found our contribution was significant (contribution scores of 1,3,3,4).

We have uploaded a revision of our manuscript (including the appendix) with changes colored in *red* for clarity. The manuscript includes:
* A clearer, more general version of our theoretical results. In our revision, we provide an updated theoretical formulation that should help clarify some of the questions raised by the reviewers. The main change compared to the initial manuscript is that the results are derived for general parameterizations of probability distributions (using compactness and continuity arguments), with the specific parameterizations described in the main text becoming special cases of those results. This in particular should help reduce confusion around the role of “discretizing” the solutions obtained in the reparameterized space.
* Comparisons with additional baseline methods: an evolutionary algorithm from NeverGrad and a baseline where discrete configurations are enumerated and continuous parameters (if any) are optimized for each discrete configuration (the gold standard, when computationally feasible)
* A more prominent presentation of the original finding that PR sees little-to-no performance degradation and large speed-ups with 128 MC samples relative to 1,024.
* A comparison of acquisition optimization performance showing that PR works best for any wall time budget on the chemistry problem.
* Improvements to the text per reviewer feedback.

---

### Meta-Review · Area_Chair_pcWu · 2022-08-26

**Recommendation:** Accept
**Confidence:** Certain

**Metareview:**

This paper studies a Bayesian optimization method where some of the variables are discrete and some are continuous and proposes using probabilistic reparameterization. Reviewers unanimously agreed that the paper is well-written, solves an important real-world problem, and most reviewers found the experimental results to be convincing. In addition, the reviewers found the rebuttal and revision to be convincing and clarified many of the initial questions.

Several reviewers pointed out that some of the results referenced in the paper could be more explicitly referenced in the paper and/or explained in the appendix. For the final version, please make an effort to address reviewers feedback to make the paper more self-contained.

**Award:**

Yes

---

### Decision · Program_Chairs · 2022-09-14

Accept